# Achieving Domain-Independent Certified Robustness via *Knowledge Continuity*

**Alan Sun**[1,2], **Chiyu Ma**[2], **Kenneth Ge**[1], **Soroush Vosoughi**[2]
[1]Carnegie Mellon University, [2]Dartmouth College
{alansun, kkge}@andrew.cmu.edu,
{chiyu.ma.gr, soroush.vosoughi}@dartmouth.edu

## Abstract

We present *knowledge continuity*, a novel definition inspired by Lipschitz continuity which aims to certify the robustness of neural networks across input domains (such as continuous and discrete domains in vision and language, respectively). Most existing approaches that seek to certify robustness, especially Lipschitz continuity, lie within the continuous domain with norm and distribution-dependent guarantees. In contrast, our proposed definition yields certification guarantees that depend only on the loss function and the intermediate learned metric spaces of the neural network. These bounds are independent of domain modality, norms, and distribution. We further demonstrate that the expressiveness of a model class is not at odds with its knowledge continuity. This implies that achieving robustness by maximizing knowledge continuity should not theoretically hinder inferential performance. Finally, to complement our theoretical results, we present several applications of knowledge continuity such as regularization, a certification algorithm, and show that knowledge continuity can be used to localize vulnerable components of a neural network[1].

## 1 Introduction

Deep neural networks (DNNs) have demonstrated remarkable generalization capabilities. Their robustness, however, has been considerably more difficult to achieve. *Robustness* refers to the preservation of model performance under natural or adversarial alterations of the input [18]. DNNs' lack of robustness, highlighted by seminal works such as [24, 66] and recently [7, 5], poses significant challenges to their adoption in critical applications, underscoring concerns for AI safety and trustworthiness [20, 30, 9, 7].

Though issues of robustness emerged from computer vision applications, they have since spanned multiple domains [1, 35, 72, 75, 7]. This research trajectory has not only prompted significant advancements in robustness improvements through architectural, training, and dataset augmentations, but also unveiled the sophistication of *adversarial attacks*—the process through which counterexamples to robustness are generated [1, 35, 72, 75, 7]. Along the progress made in these parallel directions, a great deal of work has gone into *certified robustness* which seeks to provide theoretical robustness guarantees. Certification is desirable as it generally transcends any particular task, dataset, or model.

As a result, *Lipschitz continuity* has emerged, promising certified robustness by essentially bounding the derivative of a neural network's output with respect to its input. In this way, Lipschitz continuity directly captures the volatility of a model's performance, getting at the heart of robustness. Such an approach has proven its merit in computer vision, facilitating robustness under norm and distributional assumptions [29, 59, 78, 76]. Its inherent ease and interpretability has lead to widespread adoption as a means to measure and regulate robustness among practitioners as well [71, 12, 21, 68, 54].

---

[1]Codebase for our experiments can be found at https://github.com/alansun17904/kc

38th Conference on Neural Information Processing Systems (NeurIPS 2024).

Despite these successes in computer vision, there are fundamental obstacles when one tries to apply Lipschitz continuity into discrete or non-metrizable domains such as natural language. Firstly, characterizing distance in this input-output space is highly nontrivial, as language does not have a naturally-endowed distance metric. Additionally, suppose we impose some distance metric on the input-output space [49, 16]. For such a metric to meaningfully characterize adversarial perturbations, it cannot be universally task-invariant. Consider the two sentences (a) "I am happy," (b) "I am sad." The ground-truth label of (a) is invariant to the perturbation (a) → (b), if the task is sentence-structure identification, but it would not be preserved for a task like sentiment classification. Lastly, key architectures such as the Transformer [70] are provably *not* Lipschitz continuous [36]. ***Most of these challenges are not unique to language, and they represent a strong divide of our understanding of robustness in discrete/non-metrizable and continuous domains [22, 46].***

To address these issues, we propose a new conceptual framework which we call *knowledge continuity*. At its core, we adopt the following axiom:

*Robustness is the stability of a model's performance*
*with respect to its **perceived** knowledge of input-output relations.*

Concretely, our framework is grounded on the premise that robustness is better achieved by focusing on the variability of a model's loss with respect to its hidden representations, rather than forcing arbitrary metrics on its inputs and outputs. Our approach results in certification guarantees independent of domain modality, norms, and distribution. We demonstrate that the expressiveness of a model class is not at odds with its knowledge continuity. In other words, achieving robustness by improving knowledge continuity should not theoretically hinder inferential performance. We show that in continuous settings (i.e. computer vision) knowledge continuity generalizes Lipschitz continuity and inherits its tight robustness bounds. Finally, we present an array of practical applications using knowledge continuity both as an indicator to predict and characterize robustness as well as an additional term in the loss function to train robust classifiers. In sum, our contributions are threefold:

- Introduction of *knowledge continuity*, a new concept that frames robustness as variability of a model's loss with respect to its hidden representations.

- We theoretically show that knowledge continuity results in certified robustness guarantees that generalize across modalities (continuous, discrete, and non-metrizable). Moreover, this robustness does not come at the expense of inferential performance.

- We present several practical applications of knowledge continuity such as using it train more robust models, in both language processing and vision, identify problematic hidden layers, and using its theoretical guarantees to formulate a novel certification algorithm.

Although our results apply to all discrete/non-metrizable and continuous spaces, throughout the paper we invoke examples from natural language as it culminates the aforementioned challenges. Further, the ubiquity of large language models make their robustness a timely focus.

## 2  Related Works

There have been extensive studies on developing robust neural networks with theoretical guarantees. With respect to our contributions, they can be organized into the following categories.

**Certified robustness with Lipschitz continuity.** The exploration of Lipschitz continuity as a cornerstone for improving model robustness has yielded significant insights, particularly in the domain of computer vision. This principle, which ensures bounded derivatives of the model's output with respect to its input, facilitates a smoother model behavior and inherently encourages robustness against adversarial perturbations. This methodology, initially suggested by [24], has since been rigorously analyzed and expanded upon. Most theoretical results in this area focus on certifying robustness with respect to the $\ell_2$-norm [11, 86, 25, 2, 38, 29, 4]. A recent push, fueled by new architectural developments, has also expanded these results into $\ell_\infty$-norm perturbations [89, 88, 90]. Further, Lipschitz continuity-inspired algorithms also serve practitioners as a computationally effective way to train more robust models [68, 78, 69, 13]. This stands in contrast to (virtual) adversarial training methods which brute-force the set of adversarial examples, then iteratively retrain on them [50, 63, 80]. Though Lipschitz continuity has seen much success in continuous domains, it does not apply to non-metrizable domains such as language. Further, architectural limitations of prevalent models such as

the Transformer [70, 36] exacerbate this problem. These challenges highlight a critical need for a new approach that can accommodate the specificities of discrete and non-metrizable domains while providing robustness guarantees.

**Achieving robustness in discrete/non-metrizable spaces.** Non-metrizable spaces, where it is non-trivial to construct a distance metric on the input/output domains, pose a unique challenge to certified robustness. Instead of focusing on point-wise perturbations, many studies have opted to examine how the output probability distribution of a model changes with respect to input distribution shifts by leveraging information bottleneck methods [67, 73, 53] (see also out-of-distribution generalization: [42, 83, 60]). Most of these bounds lack granularity and cannot be expressed in closed-form. In contrast to these theoretical approaches, recent efforts have refocused on directly adapting the principles underlying Lipschitz continuity to language. Virtual adversarial training methods such as [43, 85] mimic the measurement of Lipschitz continuity by comparing changes in the textual embeddings with the KL-divergence of the output logits. Along these lines, techniques akin to those used in adversarial training in vision have also been translated to language, reflecting a shift towards robustness centered around the learned representation space [40, 23, 35]. Though these approaches have seen empirical success, they lack theoretical guarantees. As a result, their implementations and success rate is heavily task-dependent [43, 85]. There have also been attempts to mitigate the non-Lipschitzness of Transformers [87, 82] by modifying its architecture. These changes, however, add significant computational overhead.

**Other robustness approaches.** In parallel, other certified robustness approaches such as randomized smoothing [12, 39, 37] give state-of-the-art certification for $\ell_2$-based perturbations. Notable works such as [34, 74] have sought to generalize these techniques into language, but their guarantees strongly depend on the type of perturbation being performed. On the other hand, analytic approaches through convex relaxation inductively bound the output of neurons in a ReLU network across layers [79, 81, 77]. These works, however, are difficult to scale and also do not transfer easily to discrete/non-metrizable domains.

Our approach, inspired by Lipschitz continuity, distills the empirical intuitions from the works of [43, 85] and provides theoretical certification guarantees independent of perturbation-type [34, 74] and domain modality. We demonstrate that knowledge continuity yields many practical applications analogous to Lipschitz continuity which are easy to implement and are computationally competitive.

## 3 Preliminaries

**Notations.** Let $\mathbb{R}^{\geq 0} := [0, \infty)$. For any function $f : \mathcal{X} \to \mathcal{Y}$, we denote $\mathrm{graph}(f) := \{(x, y) \in \mathcal{X} \times \mathcal{Y} : f(x) = y\}$. For $n \in \mathbb{N}$, let $[n]$ denote the set $\{1, 2, \ldots, n\}$. $(\mathcal{X}, \mathcal{F}_\mathcal{X}, \mathbb{P}_\mathcal{X})$, $(\mathcal{Y}, \mathcal{F}_\mathcal{Y}, \mathbb{P}_\mathcal{Y})$ are probability spaces and $(\mathcal{X} \times \mathcal{Y}, \mathcal{F}_\mathcal{X} \otimes \mathcal{F}_\mathcal{Y}, \mathbb{P}_\mathcal{X} \times \mathbb{P}_\mathcal{Y})$ denotes the product measurable space of the probability spaces $\mathcal{X}, \mathcal{Y}$. Since our contribution focuses on the supervised learning regime, we colloquially refer to $\mathcal{X}, \mathcal{Y}$ as the input and labels, respectively. We call any probability measure $\mathbb{P}_{\mathcal{X} \times \mathcal{Y}}$ absolutely continuous to $\mathbb{P}_\mathcal{X} \times \mathbb{P}_\mathcal{Y}$ (i.e. $(\mathbb{P}_\mathcal{X} \times \mathbb{P}_\mathcal{Y})(E) = 0 \Rightarrow \mathbb{P}_{\mathcal{X} \times \mathcal{Y}}(E) = 0$) a *data distribution* and denote it as $\mathcal{D}_{\mathcal{X}, \mathcal{Y}}$. If $(\mathcal{Z}, d_\mathcal{Z})$ is a metric space with metric $d_\mathcal{Z}$ and $A \subset \mathcal{Z}$, then for any $z \in \mathcal{Z}$, $d_\mathcal{Z}(z, A) = \inf_{a \in A} d_\mathcal{Z}(a, z)$. We say that a metric space, $(\mathcal{Z}, d_\mathcal{Z})$, is bounded by some $B \in \mathbb{R}^{\geq 0}$, if $\sup_{z', z \in \mathcal{Z}} d(z, z') < B$. Denote by $\mathrm{Id}_\mathcal{Z} : \mathcal{Z} \to \mathcal{Z}$ the identity function on $\mathcal{Z}$. Let $\mathcal{L} : \mathcal{Y} \times \mathcal{Y} \to \mathbb{R}^{\geq 0}$ be a loss function where $\mathcal{L}(y, y') = 0$ if and only if $y = y'$. For any $f : \mathcal{X} \to \mathcal{Y}$ and $(x, y), (x', y') \in \mathcal{X} \times \mathcal{Y}$, we denote $\Delta \mathcal{L}_f^{(x,y)}(x', y') := |\mathcal{L}(f(x), y) - \mathcal{L}(f(x'), y')|$, essentially the absolute difference in loss between $(x, y)$ and $(x', y')$. Unless otherwise specified, it will be assumed that $f$ is a measurable function from $\mathcal{X}$ to $\mathcal{Y}$ with a metric decomposition (see Def. 1).

**Lipschitz continuity.** Given two metric spaces $(\mathcal{X}, d_\mathcal{X}), (\mathcal{Y}, d_\mathcal{Y})$ a function $f : \mathcal{X} \to \mathcal{Y}$ is $K$-Lipschitz continuous if there exists $K \in \mathbb{R}^{\geq 0}$ such that for all $x, x' \in \mathcal{X}$, $d_\mathcal{Y}(f(x), f(x')) \leq K d_\mathcal{X}(x, x')$.

## 4 Knowledge Continuity

In this section, we provide the formal definition of *knowlege continuity* and explore its theoretical properties.

We start by defining a model's perceived knowledge through a rigorous treatment of its hidden representation spaces. By considering the distance between inputs in some representation space in conjunction with changes in loss, we result in a measure of *volatility* analogous to Lipschitz continuity.

Bounding this volatility in expectation then directly leads to our notion of knowledge continuity. With these tools, we demonstrate a host of theoretical properties of knowledge continuity including its certification of robustness, guarantees of expressiveness, and connections to Lipschitz continuity in continuous settings. We summarize our theoretical contributions as follows:

- We **define** the perceived knowledge of a model as well as volatility and knowledge continuity within a model's representation space (see Def. 1, 2, 3, 4, respectively).

- We **prove** that knowledge continuity implies *probabilistic* certified robustness under perturbations in the representation space and constraining knowledge continuity should not hinder the expressiveness of the class of neural networks (see Thm. 4.1 and Prop. 4.3, 4.4, respectively).

- We **prove** that in some cases knowledge continuity is equivalent (in expectation) to Lipschitz continuity. This shows that our axiomization of robustness aligns with existing results when perturbation with respect to the input is well-defined (see Prop. 4.6, 4.8).

### 4.1 Defining Perceived Knowledge

Knowledge is generally understood as a relational concept: it arises from the connections we make between ideas, experiences, and stimuli [26]. Herein, we capture the *perceived knowledge* of a model by focusing on the relations it assigns to input-input pairs. Specifically, these relations are exposed by decomposing a function $f : \mathcal{X} \to \mathcal{Y}$ into projections to intermediate metric spaces. Formally,

**Definition 1** (Metric Decomposition). *We say that $f$ admits a metric decomposition if there exists metric spaces $(\mathcal{Z}_1, d_1), \ldots, (\mathcal{Z}_n, d_n)$ with metrics $d_k$ for $k \in [n]$ such that*

1. *$(\mathcal{Z}_k, d_k)$ is endowed with its Borel $\sigma$-algebra.*
2. *There exists measurable mappings $h_0, h_1, \ldots, h_n$ where $h_0 : \mathcal{X} \to \mathcal{Z}_1$, $h_k : \mathcal{Z}_k \to \mathcal{Z}_{k+1}$ for $k \in [n-1]$, and $h_n : \mathcal{Z}_n \to \mathcal{Y}$.*
3. *$f = h_n \circ h_{n-1} \circ \ldots \circ h_1 \circ h_0$.*

*Remark* 1. If $\mathcal{X}$ is a metric space with metric $d_{\mathcal{X}}$ and $\mathcal{F}_{\mathcal{X}}$ is its Borel $\sigma$-algebra, then for any measurable mapping $f : \mathcal{X} \to \mathcal{Y}$ there exists the trivial metric decomposition

$$f = f \circ \mathrm{Id}_{\mathcal{X}}. \tag{4.1}$$

Therefore, in computer vision applications where $(\mathcal{X}, d_{\mathcal{X}}) = (\mathbb{R}^n, \ell_p)$ for some $n \in \mathbb{Z}^+$, we can apply this trivial decomposition to yield bounds which mirror the certification guarantees of Lipschitz continuity. This is discussed in detail in Section 4.5.

To the best of our knowledge, all deep learning architectures admit metric decompositions, since their activations are generally real-valued. So, for all subsequent functions from $\mathcal{X}$ to $\mathcal{Y}$, unless otherwise specified, we assume they are measurable and possess a metric decomposition. Further, we denote $f^k = h_k \circ h_{k-1} \circ \ldots \circ h_1 \circ h_0$ and adopt the convention of calling $h_k$ the $k^{\text{th}}$ hidden layer. In Appendix A, we present several metric decompositions for a variety of architectures.

For any metric-decomposible function, an immediate consequence of our definition is that its metric decomposition may not be unique. However, in the context of neural networks, this is a desirable property. Seminal works from an array of deep learning subfields such as semi-supervised learning [57], manifold learning [51], and interpretability [10, 47] place great emphasis on the quality of learned representation spaces by examining the induced-topology of their metrics. This often does not affect the typical performance of the estimator, but has strong robustness implications [33]. Our results, which are dependent on particular metric decompositions, capture this trend. In Section 4.4, we discuss in detail the effects of various metric decompositions on our theoretical results.

### 4.2 Defining Knowledge Continuity

We first introduce what it means for a model's performance to be volatile at a data point relative to its metric decomposition. Then, we contrast knowledge continuity with Lipschitz continuity, pointing out key differences that will allow us to prove more general bounds.

**Definition 2** (k-Volatility). *Let $f : \mathcal{X} \to \mathcal{Y}$ and $\mathcal{L}$ be any loss function. The k-volatility of a point $(x, y) \in \mathcal{X} \times \mathcal{Y}$ which we denote as $\sigma_f^k(x, y)$ is given by*

$$\sigma_f^k(x, y) := \mathbb{E}_{\substack{(x', y') \sim \mathcal{D}_{\mathcal{X}, \mathcal{Y}} \\ f(x) \neq f(x')}} \left[ \frac{\Delta \mathcal{L}_f^{(x,y)}(x', y')}{d_k(f^k(x), f^k(x'))} \right], \tag{4.2}$$

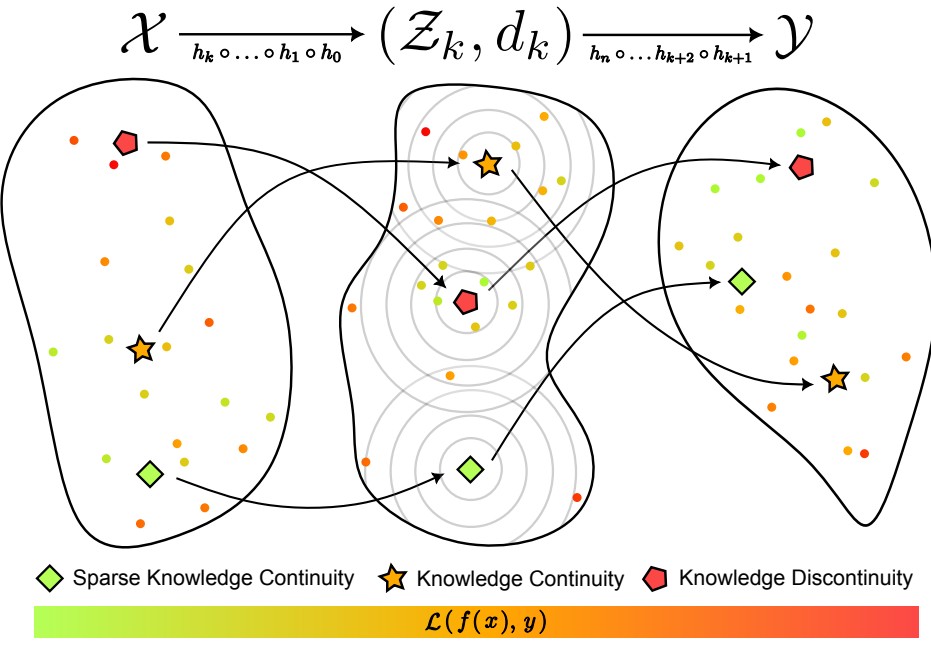

$$\mathcal{X} \xrightarrow[h_k \circ \ldots \circ h_1 \circ h_0]{} (\mathcal{Z}_k, d_k) \xrightarrow[h_n \circ \ldots h_{k+2} \circ h_{k+1}]{} \mathcal{Y}$$

◇ Sparse Knowledge Continuity ★ Knowledge Continuity ⬠ Knowledge Discontinuity

$$\mathcal{L}(f(x), y)$$

Figure 1: Examples of knowledge (dis)continuities. $f : \mathcal{X} \to \mathcal{Y}$ is a measurable map, and $(\mathcal{Z}_k, d_k)$ is one of its hidden representations. The color of the points indicates loss. ◆ denotes knowledge continuity induced by sparsity: an isolated concept with no knowledge relations close to it. So, any perturbation moves ◆ far away with high probability. Smooth changes in loss around ★ implies knowledge continuity. Finally, ⬟ is not knowledge continuous due to drastic changes in loss nearby. Notice that the classification of points is independent of input/output clustering behavior since $\mathcal{X}, \mathcal{Y}$ may not be endowed with a metric.

*where $d_k$ is the distance metric associated with $f$'s $k^{th}$ hidden layer.*

By performing some algebra on the definition, we see that it decomposes nicely into two distinct terms: *sparsity* of the representation and *variation* in loss.

$$\sigma_f^k(x, y) = \mathbb{E}_{(x', y') \sim \mathcal{D}_{\mathcal{X}, \mathcal{Y}}} \left[ \frac{|\mathcal{L}(f(x), y) - \mathcal{L}(f(x'), y')|}{d_k(f^k(x), f^k(x'))} \right],$$

$$= \mathcal{L}(f(x), y) \, \mathbb{E}_{(x', y') \sim \mathcal{D}_{\mathcal{X}, \mathcal{Y}}} \left[ \underbrace{\frac{1}{d_k(f^k(x), f^k(x'))}}_{\text{sparsity}} \cdot \underbrace{\left| 1 - \frac{\mathcal{L}(f(x'), y')}{\mathcal{L}(f(x), y)} \right|}_{\text{variation in loss}} \right], \qquad (4.3)$$

Our notion of volatility essentially measures the change in performance with respect to perturbations to a model's perceived knowledge. In particular, Eq. 4.3 reveals that there are two interactions in play which we illustrate in Fig. 1. Informally, we say that $(x, y)$ is highly volatile if there is a large discrepancy in performance between it and points that are perceived to be conceptually similar. Therefore, highly volatile points capture inaccurate input-input knowledge relations. Additionally, $(x, y)$ experiences low volatility if the space around it is sparse with respect to $\mathcal{D}_{\mathcal{X}, \mathcal{Y}}$. In other words, any set of perturbations applied in $\mathcal{Z}_k$ would push $(x, y)$ far away, with high probability. This makes $(x, y)$ an isolated concept with little knowledge relationships associated with it.

Similar to Lipschitz continuity, the boundedness of the $k$-volatility of $f$ across the data distribution is crucial and we denote this class of functions as *knowledge continuous*.

**Definition 3** (Pointwise $\epsilon$-Knowledge Continuity). *We say that $f$ is $\epsilon$-knowledge continuous at $(x, y) \in \mathcal{X} \times \mathcal{Y}$ with respect to a function $f$, loss function $\mathcal{L}$, and hidden layer $k$ if $\sigma_f^k(x, y) < \epsilon$.*

Conversely, we say that $(x, y)$ is $\epsilon$-knowledge discontinuous if the previous inequality does not hold. Further, $(x, y)$ is simply knowledge discontinuous if $\sigma_f^k(x, y)$ is unbounded. Now, we extend this definition globally by considering the $k$-volatility between all pairs of points.

**Definition 4** (Expected $\epsilon$-Knowledge Continuity)**.** *We say that $f$ is $\epsilon$-knowledge continuous with respect to a loss function $\mathcal{L}$ and hidden layer $k$ if*

$$\mathbb{E}_{(x,y)\sim\mathcal{D}}[\sigma_f^k(x,y)] < \epsilon. \tag{4.4}$$

Though the functional forms of Lipschitz continuity and knowledge continuity are similar, there are important differences that allow us to prove more general results. Firstly, ***unlike Lipschitz continuity which is an analytical property of the model $f$, knowledge continuity is a statistical one.*** In this way, non-typical data points, even if they are volatile, are ignored, whereas Lipschitz continuity treats all points equally. This is necessary in many discrete applications, as projecting a countable input space onto a non-countable metric space inevitably results in a lack of correspondence thereof. Moreover, ground-truth relations from $\mathcal{X} \to \mathcal{Y}$ may not be well-defined on *all* of $\mathcal{X}$: consider sentiment classification of an alpha-numeric UUID string or dog-cat classification of Gaussian noise. Secondly, ***the knowledge continuity of an estimator is measured with respect to the loss function rather than its output.*** This property allows us to achieve the expressiveness guarantees in Section 4.4, since it places no restrictions on the function class of estimators. Lastly, ***knowledge continuity measures the distance between inputs with the endowed metric in its hidden layers.*** This flexibility allows us to define knowledge continuity even when the input domain is not a metric space.

## 4.3 Certification of Robustness

Our first main result demonstrates that $\epsilon$-knowledge continuity implies *probabilistic* certified robustness in the hidden representation space. In Theorem 4.1, given some reference set $A \subset \mathcal{X} \times \mathcal{Y}$, we bound the probability that a $\delta$-sized perturbation in the representation space away from $A$ will result in an expected $\eta$ change in loss. In other words, knowledge continuity is able to characterize the robustness of any subset of data points with positive measure.

**Theorem 4.1.** *Let $A \subset \mathcal{X} \times \mathcal{Y}$ such that $\mathbb{P}_{D_{\mathcal{X},\mathcal{Y}}}[A] > 0$ and $\delta, \eta > 0$. Let $A' = \left\{ (x', y') \in \mathcal{X} \times \mathcal{Y} : \mathbb{E}_{\substack{(x,y)\sim D_{\mathcal{X},\mathcal{Y}} \\ (x,y)\in A}} \Delta \mathcal{L}_f^{(x,y)}(x', y') > \eta \right\}$. If $f : \mathcal{X} \to \mathcal{Y}$ is $\epsilon$-knowledge continuous with respect to the hidden layer indexed by $k$ and $(\mathcal{Z}_k, d_k)$ is bounded by $B > 0$, then*

$$\mathbb{P}_{(x,y)\sim D_{\mathcal{X},\mathcal{Y}}}[A' \mid d_k(f^k(x), f^k(A)) < \delta] \le \frac{\epsilon\delta}{\eta\left(1 - \exp\left[-\Omega\left(\frac{\delta}{B} - \sqrt{\log\frac{1}{\mathbb{P}[A]}}\right)^2\right]\right)}. \tag{4.5}$$

*Proof sketch.* We apply the definition of conditional probability $P(A|B) = P(A \cap B)/P(B)$ and bound $P(A \cap B)$, $P(B)$, separately. The numerator, $\mathbb{P}[A'$ and $d_k(f^k(x), f^k(A)) < \delta]$, is upper-bounded through an application of Markov's Inequality. On the other hand, we apply known concentration inequalities to lower bound $\mathbb{P}[d_k(f^k(x), f^k(A)) < \delta]$, combining these results in the theorem. We present the proof in its entirety in Appendix B. ∎

This demonstrates that knowledge continuity results in certification of robustness, independent of distance metric and domain modality. The assumption of boundedness and requirement to know $\mathbb{P}[A]$ can be lost by taking limits of Eq. 4.5 with respect to $B$ and $\mathbb{P}[A]$. This yields the following corollary.

**Corollary 4.2.** *If $(\mathcal{Z}_k, d_k)$ is unbounded, then*

$$\mathbb{P}_{(x,y)\sim D_{\mathcal{X},\mathcal{Y}}}[A' \mid d_k(f^k(x), f^k(A)) < \delta] \le \frac{\epsilon\delta}{\eta(1 - \mathbb{P}[A])}. \tag{4.6}$$

*If $\mathbb{P}[A] = 0$, then*

$$\mathbb{P}_{(x,y)\sim D_{\mathcal{X},\mathcal{Y}}}[A' \mid d_k(f^k(x), f^k(A)) < \delta] \le \frac{\epsilon\delta}{\eta}. \tag{4.7}$$

*Proof.* These results follow from directly taking the limit as $B \to \infty$ and applying some of the bounds acquired in the proof of Thm. 4.1. This yields Eq. 4.6. Next, jointly taking the limit as $\mathbb{P}[A] \to 0$ and $B \to \infty$ results in Eq. 4.7. ∎

In both Thm. 4.1 and Cor. 4.7, we yield probabilistic guarantees like [12], rather than deterministic ones. Though deterministic bounds are desirable, the stochasticity of our framework is necessary

for its generalization across different domains. For most continuous, metrizable applications (like computer vision), models learn a hidden representation space where most minute changes in this space correspond to tangible inputs. The same cannot be said for many discrete or non-metrizable applications. In natural language processing, the correspondence between the learned representation space and the input is sparse, resulting in lots of "dead space": portions of the hidden representation space that do not correspond to any input [3, 19]. And so, by incorporating the data distribution into our bounds, we implicitly adjust for this: assigning zero-measure to the aforementioned "dead space."

## 4.4 Expressiveness

Our second main result demonstrates that $\epsilon$-knowledge continuity can be achieved without theoretically compromising the accuracy of the model. In other words, universal function approximation is an invariant property with respect to $\epsilon$-knowledge continuity. Universal approximation results have seen a great deal of theoretical work, as they put limits on what neural networks can represent [15, 31, 45]. As discussed in Section 2, Lipschitz continuous functions do not achieve universal function approximation with respect to the set of all functions, in particular, non-continuous ones. However, we show that under strong conditions this is achievable with knowledge continuity.

First, let us formally define a *universal function approximator*.

**Definition 5** (Universal Function Approximator). *Suppose that $\mathcal{L}$ is Lebesgue-integrable in both coordinates. Let $\mathcal{F} \subset \mathcal{Y}^{\mathcal{X}}$ be a set of measurable functions from $\mathcal{X} \to \mathcal{Y}$ such that for any $f \in \mathcal{F}$, there exists $\mu_f \ll \mathcal{D}_{\mathcal{X},\mathcal{Y}}$ such that $\mu_f(graph(f)) = 1$. Then, $\mathcal{U} \subset \mathcal{F}$ is a universal function approximator of $\mathcal{F}$ if for every $f \in \mathcal{F}$ and every $\epsilon > 0$, there exists $\hat{f} \in \mathcal{U}$ such that*

$$\int \mathcal{L}(\hat{f}(x), y) \, d\mu_f < \epsilon. \tag{4.8}$$

We now show any universal function approximator can be made robust through the trivial metric decomposition.

**Proposition 4.3.** *Let $\mathcal{U} \subset \mathcal{Y}^{\mathcal{X}}$ be a universal function approximator of $\mathcal{Y}^{\mathcal{X}}$ with respect to some loss function $\mathcal{L}$. Then, for any $f \in \mathcal{Y}^{\mathcal{X}}$ and sequence $\epsilon_1, \epsilon_2, \ldots$ such that $\epsilon_n \to 0$ there are a sequence of $\epsilon_n$-knowledge continuous functions in $\mathcal{U}$ such that $\int \mathcal{L}(f_n(x), y) \, d\mu_f < \epsilon_n$, for $n \in \mathbb{N}$.*

*Proof.* Choose $f_n \in \mathcal{U}$ such that $\int \mathcal{L}(f_n(x), y) \, d\mu_f < \frac{1}{2}\epsilon_n$. Consider the 1-layer metric decomposition of $f$, $h_1 : \mathcal{X} \to \mathcal{Z}_1$ where $\mathcal{Z}_1 = \mathcal{X}$ equipped with the trivial metric ($d_1(x, y) = 1$ if $x \neq y$ and 0 otherwise). Then, $f_n = f_n \circ h_1$. So, it follows that

$$\mathbb{E}\,\sigma_{f_n}^1(x, y) = \int \frac{\Delta \mathcal{L}_{f_n}^{(x,y)}(x', y')}{d_1(h_1(x), h_1(x'))} \, d\mu_f,$$

$$\leq \int \Delta \mathcal{L}_{f_n}^{(x,y)}(x', y') \, d\mu_f,$$

$$\leq \epsilon_n.$$

and by the construction of $f_n$, the proof is completed. ∎

In other words, if our estimator was given "infinite representational capacity," robustness can be trivially achieved by isolating every point as its own concept (as discussed in Section 4.2). More generally, if we instead considered a generalized discrete metric (fix $c \in [0, \infty]$, $d(x, y) = c$ if and only if $x = y$ and $d(x, y) = 0$, otherwise), then as $c \to \infty$, $k$-volatility converges pointwise to 0 almost everywhere assuming that the loss is finite almost everywhere. In practice, we find these degenerate decompositions to be unreasonable as they also trivialize robustness. For example, if $c = \infty$, then robustness is not well-defined as any perturbation would lead to a point that is perceived to be infinitely far away. ***In this sense, our framework accounts for different notions of robustness, strong and weak.*** The next result builds on Prop. 4.3 and demonstrates how a stronger notion of robustness will affect expressiveness. These added constraints make it so that trivial metric decompositions are no longer possible unless the metric in $\mathcal{X}$ is also trivial. We state this formally below, note the highlighted differences between this and Prop. 4.3.

**Proposition 4.4.** *Suppose $(\mathcal{X}, d_{\mathcal{X}}), (\mathcal{Y}, d_{\mathcal{Y}}) := (\mathcal{X}, d_{\mathcal{X}})$ are **compact** metric spaces, $\mathcal{F} \subset \mathcal{Y}^{\mathcal{X}}$ is the **set of all continuous functions** from $\mathcal{X}$ to $\mathcal{Y}$ such that $\int d_{\mathcal{X}}(x, x')^{-1} d\mu_f < \infty$ and $\mathcal{L}$ be Lipschitz continuous in both coordinates. Then, there exists a universal function approximator $\mathcal{U}$ of $\mathcal{F}$ that is knowledge continuous (i.e. $\mathbb{E}\,\sigma_f^k(x, y) < \infty$ for some $k$).*

*Proof sketch.* We show an outline of the proof here and defer the full proof to Appendix C. By the Stone-Weierstrass Theorem, the set of Lipschitz continuous functions is dense in the set of all continuous functions from $\mathcal{X}$ to $\mathcal{Y}$. Since $\mathcal{L}$ is Lipschitz continuous in both coordinates, through some algebra, $\mathbb{E}\,\sigma_f^1(x, y) < \infty$, where $h_1 = \text{Id}_\mathcal{X}$ and we yield the statement of the theorem. ∎

The additional constraint $\int d_\mathcal{X}(x, x')^{-1} d\mu_f$ requires data points to be sparsely layed out in the representation space. As discussed previously, this assumption is generally reasonable for discrete applications. In conjunction with Prop. 4.3, we have shown that the class of knowledge continuous functions is ***strictly larger*** than the class of Lipschitz continuous ones. Though we show that universal approximation by knowledge continuous networks is achievable, it is unclear whether these results still hold if the "tightness" of the metric decompositions is bounded. Specifically, the construction in Prop. 4.3 results in a metric decomposition with infinite Hausdorff dimension. Is it possible to achieve Prop. 4.3 in its most general form if we only consider the set of all knowledge continuous functions with metric decompositions with finite Hausdorff dimension? Based on the theoretical and empirical results of [62, 33], respectively, we conjecture in the negative and leave its resolution open.

**Conjecture 4.5.** *If $\mathcal{U} \subset \mathcal{Y}^\mathcal{X}$ is a universal function approximator with respect to some Lebesgue-integrable loss function $\mathcal{L}$. Then, for any $f \in \mathcal{Y}^\mathcal{X}$, there **does not exist** a sequence of functions with metric decompositions of **finite Hausdorff dimension** that achieve arbitrarily small approximation error (i.e. $\int \mathcal{L}(\hat{f}(x), y) d\mu_f$) and knowledge continuity.*

### 4.5 Connections to Lipschitz Continuity

We now demonstrate that our axiomization of robustness presented in Section 1 aligns with the notion of robustness[2] commonly prescribed in vision [18]. This unifies the certified robustness bounds with respect to the representation space derived in Thm. 4.1 with existing work certifying robustness with respect to the input space in continuous applications such as vision.

Our first result identifies conditions under which knowledge continuity, implies Lipschitz continuity.

**Proposition 4.6.** *Suppose that $(\mathcal{X}, d_\mathcal{X}), (\mathcal{Y}, d_\mathcal{Y})$ are metric spaces. Let the first $n$ metric decompositions of $f : \mathcal{X} \to \mathcal{Y}$ be $K_i$-Lipschitz continuous, for $i \in [n]$. If $f$ is $\epsilon$-knowledge continuous with respect to the $n^{th}$ hidden layer and $d_\mathcal{Y}(f(x), f(x')) \leq \eta \Delta \mathcal{L}_f^{(x,y)}(x', y)$ for all $x, x' \in \mathcal{X}$, $y \in \mathcal{Y}$, and some $\eta > 0$, then $f$ is Lipschitz continuous in expectation. That is,*

$$\mathbb{E}_{(x,y),(x',y') \sim \mathcal{D}_{\mathcal{X},\mathcal{Y}}} \frac{d_\mathcal{Y}(f(x), f(x'))}{d_\mathcal{X}(x, x')} \leq \epsilon \eta \prod_{j=1}^n K_j. \tag{4.9}$$

The proof is presented in Appendix D and follows easily through some algebraic manipulation. It is easy to see that if $f$ is knowledge continuous with respect to some identity (or contractive) metric decomposition, then we can loose the repeated product. Analogous to Remark 1, the concepts of Lipschitz continuity and knowledge continuity become similar when we can assign metrics to the input-output spaces. Next, combining this proposition with an auxiliary result from [89], we directly yield a certification on the input space.

**Corollary 4.7.** *Suppose that assumptions of Prop. 4.6 are true. And also assume that $(\mathcal{X}, d_\mathcal{X}) = (\mathbb{R}^n, \ell_p), (\mathcal{Y}, d_\mathcal{Y}) = (\mathbb{R}^m, \ell_p)$, for $1 \leq p \leq \infty$. Define a classifier from $f : \mathbb{R}^n \to \mathbb{R}^m$, $g$, where $g(x) := \arg\max_{k \in [m]} f_k(x)$ for any $x \in \mathbb{R}^n$. Then, with probability $1 - \frac{\epsilon \eta}{t} \prod_{j=1}^n K_j$, $g(x) = g(x + \delta)$ for all $\|\delta\|_p < (2^{1/p}/2t) margin(f(x))$ and $t > 0$. $f_k(x)$ is the $k^{th}$ coordinate of $f(x)$ and $margin(f(x))$ denotes the difference between the largest and second-largest output logits.*

We present the proof in Appendix D. Our second result identifies conditions under which Lipschitz continuity, implies knowledge continuity.

**Proposition 4.8.** *Let $(\mathcal{X}, d_\mathcal{X}), (\mathcal{Y}, d_\mathcal{Y})$ be a metric spaces. Let $f : \mathcal{X} \to \mathcal{Y}$ be $\epsilon$-Lipschitz continuous and $\mathcal{L}(f(x), y)$ be $\eta$-Lipschitz continuous with respect to both coordinates. If the first $n$ metric decompositions of $f$ are $K_i$-Lipschitz continuous, then $f$ is knowledge continuous with respect to the $n^{th}$ hidden layer. That is,*

$$\mathbb{E}_{(x,y) \sim \mathcal{D}_{\mathcal{X},\mathcal{Y}}} \sigma_f^n(x, y) \leq \epsilon \eta \prod_{j=1}^n \frac{1}{K_j}. \tag{4.10}$$

---

[2]Small perturbations on the input result in small changes in performance which implies small changes in output when the loss function is Lipschitz continuous.

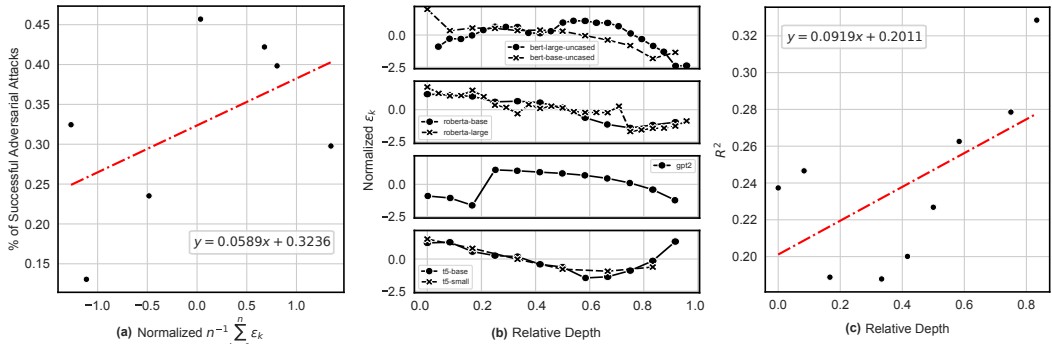

Figure 2: (a) The average percentage of successful adversarial attacks by TextFooler [35] on a host of models [58, 57, 16, 44] and the IMDB [48] dataset regressed with the average of knowledge continuity coefficients across all hidden layers ($R^2 = 0.35$). (b) $k$-Volatility as $k$ is varied across a model's relative depth. (c) Correlation between $k$-volatility and adversarial vulnerability (averaged across all models shown in (b)) with respect to TextFooler [35] as $k$ varies.

We detail the proof of this proposition in Appendix D. We note that in continuous applications such as computer vision, the assumptions of both propositions are generally met (i.e. our input-output spaces are metric spaces, all hidden layers are Lipschitz, and loss functions are locally Lipschitz). Furthermore, common architectures such as fully connected networks, CNNs, RNNs, and even vision transformers are Lipschitz continuous [71, 55]. ***This implies that our notion of robustness is indeed an appropriate generalization that transcends domain modality since in continuous settings we can recover the strong bounds of Lipschitz continuity while expanding into new discrete and non-metrizable territory.***

## 5 Practical Applications

In addition to the theoretical guarantees given by knowledge continuity in Section 4, we also demonstrate that knowledge continuity can be easily applied in practice. First, we find that knowledge continuity, similar to Lipschitz continuity, can be used to gauge adversarial robustness. Along these lines, our measure of volatility (see Def. 2) can be used to isolate particularly vulnerable hidden representations. These applications then directly motivate regulation of knowledge continuity as a means to enforce robustness.

Unless otherwise specified, we run all of our experiments on the IMDB dataset [48] (a sentiment classification task) using a host of language models from different model families (encoder, decoder, encoder-decoder). We also present additional experiments on vision tasks. These experiments can be found in the Appendix G.

**Knowledge continuity can predict adversarial robustness.** For a given model, $f$, with $n$ hidden representations, choose some $k \in [n]$. Then, consider the hidden representation index by $k$. For this fixed $k$, we determine its $k$-volatility by directly estimating Def. 2 through a naive Monte-Carlo algorithm (see Appendix G for more details). Repeating this for all $k \in [n]$, we yield a collection of $k$-volatilities which we denote as $\{\epsilon_1, \ldots, \epsilon_n\}$, one for each hidden layer. When we regress a simple average of these coefficients, $n^{-1} \sum_{k=1}^{n} \epsilon_k$, with the empirical adversarial robustness (estimated using TextFooler [35]), a strong correlation is observed. This is shown in Fig. 2(a). In particular, knowledge continuity alone is able to explain 35% of the variance in adversarial attack success rate. When we combine $k$-volatility with other model properties like size, model family, even more variance can be explained ($R^2 = 0.48$). Thus, knowledge continuity may be used as a computationally efficient method to estimate adversarial vulnerability with respect to the input space as compared to iteratively applying real adversarial attacks. Moreover, when the adversary is unknown *a priori*, knowledge continuity can also be used in this way as a diagnostic tool. A detailed discussion of these experiments are presented in Appendix E.

**Knowledge continuity can localize vulnerable hidden representations.** We plot the relationship between the $k$-volatility, $\epsilon_k$, and the relative depth of the model (i.e. $k/n$). We find that language models belonging to different model families (encoder, decoder, encoder-decoder) admit different $k$-volatility trajectories. This is shown in Fig. 2(b). In this way, knowledge continuity may provide a more

Table 1: Comparison of our knowledge continuity algorithm to existing works across various model families and adversarial attack methods. TF, BA, ANLI denote adversarial attacks [35], [40], and [52], respectively. Regulating knowledge continuity to improve robustness is superior across almost all tasks and attacks.

| Arch. | Method | IMDB | IMDB$_{TF}$ | IMDB$_{BA}$ | ANLI$_{R1}$ | ANLI$_{R2}$ | ANLI$_{R3}$ |
|---|---|---|---|---|---|---|---|
| BERT [16] ~110M params | Base | 93.6 | 47.9 | 45.2 | 44.5 | 45.6 | 33.8 |
| | TF [35] | 93.3 | 69.2 | 62.5 | ✗ | ✗ | ✗ |
| | ALUM [43] | 93.5 | 56.9 | 47.8 | 45.2 | 46.7 | **46.3** |
| | **KCReg (ours)** | **94.8** | **75.1** | **84.9** | **45.6** | **46.9** | 45.3 |
| GPT2 [57] ~1.5B params | Base | 93.6 | 63.9 | 54.9 | 42.7 | 44.9 | 43.4 |
| | TF [35] | 92.0 | 64.5 | 51.3 | ✗ | ✗ | ✗ |
| | ALUM [43] | 94.9 | 49.4 | 27.5 | 43.8 | 45.2 | 44.6 |
| | **KCReg (ours)** | **94.9** | **87.8** | **90.6** | **47.1** | **48.1** | **44.7** |
| T5 [58] ~220M params | Base | 93.7 | 53.9 | 39.3 | 46.1 | 44.7 | **46.0** |
| | TF [35] | **96.8** | 77.8 | 60.6 | ✗ | ✗ | ✗ |
| | ALUM [43] | 95.1 | 67.1 | 51.9 | 44.5 | 44.8 | 44.4 |
| | **KCReg (ours)** | 94.9 | **89.3** | **91.3** | **48.2** | **45.0** | 44.3 |

nuanced picture of a model's inductive biases and robustness beyond a scalar value like "accuracy under adversarial attack." We present a detailed analysis of this in Appendix F. Further, these dynamics may act as a diagnostic tool and offer a starting point for designing *model-specific* robustness interventions or adversarial defenses. For example, when insights from Fig. 2(b) are combined with a knowledge continuity regularization algorithm, this yields superior empirical robustness compared to existing methods. This is shown in the next subsection and in Appendix G. In addition, knowledge continuity can also quantitatively characterize an adversarial attack against a host of models which is useful for online or adaptive defenses [84, 64, 14]. This is shown in in Fig. 2(c), where TextFooler [35] largely exploits the knowledge continuities in middle/final layers of the model to decrease performance.

**Regulating knowledge continuity.** Motivated by the theoretical results in Section 4, we augment the loss function during training to mitigate knowledge continuity. Specifically, on each training iteration (batch), we start by choosing a hidden layer at random according to a Beta distribution determined *a priori*: $X \sim \text{Beta}(\alpha, \beta)$ and let $k = \lfloor nX \rfloor$. Here, $\alpha, \beta$ are chosen according to Fig. 2(b,c). We assign larger sampling probability to layers where both $k$-volatility is high and where knowledge continuity is highly correlated with adversarial robustness. In this way, our regularization objective is both model and attack specific (if the attack method is unknown, then we only apply the former). Then, we devise a Monte-Carlo algorithm to estimate this layer's $k$-volatility, $\epsilon_k$, (see Appendix G) on this minibatch. And so, the augmented loss function becomes $\mathcal{L}'(f(x), y) = \mathcal{L}(f(x), y) + \lambda \epsilon_k$ with $\lambda \geq 0$ as a hyperparameter, controlling the regularization strength. In contrast to existing adversarial training methods that perform inner-optimization steps [50, 43, 85], our method requires only additional zeroth-order computations. As a result, it outperforms existing works in training speed (up to 2× for TextFooler [35] and 3× for ALUM [43]), while improving robustness. We present a discussion of the results, ablation studies, and training details in Appendix G.

**Certifying robustness with knowledge continuity.** We present an algorithm based on Thm. 4.1 to certify robustness during test-time. Similar to [12], we estimate the probability of there existing an adversarial example within some fixed radius (in the representation space, according to a pre-defend distance metric) through bootstrapping a one-side confidence interval. Applying these methods to our regularization results, we show that regularizing knowledge continuity increases the certified robustness. The certification algorithm, its proof of correctness, and certifications of our regularized models are presented in Appendix H.

# 6 Conclusion

In this paper, we propose a novel definition, *knowledge continuity*, which addresses some of the key limitations of Lipschitz robustness. We demonstrate that our definition certifies robustness across domain modality, distribution, and norms. We also show that knowledge continuity, in contrast to Lipschitz continuity, does not affect the universal approximation property of neural networks. We also establish conditions under which knowledge continuity and Lipschitz continuity are equivalent. Lastly, we present several practical applications that directly benefit the practitioner. The broader impacts, reproducibility, and limitations of our work can be found in Appendix I, J, K, respectively.

## 7 Acknowledgements

Alan Sun thanks Fengwen Sun for the helpful feedback on early drafts of the work as well as Jeffrey Jiang and Andrew Koulogeorge for thoughtful discussions.

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

# Table of Contents

# A  More on Metric Decompositions

In Section 4.1, we introduced the notion of a *metric decomposition* to rigorously define the hidden representations of a neural network. Herein, we show that our notion of a metric decomposition well-describes a host of neural architectures and also point to possible applications of this concept beyond just deep learning. Let us first consider possible metric decompositions of common neural architectures.

## A.1  Metric Decompositions of Common Neural Architectures

**Fully-Connected Neural Network.** Suppose that $f : \mathbb{R}^d \to \mathbb{R}^m$ is a fully-connected neural network with $n$ hidden layers. Each hidden layer indexed by $i \in [n]$ has a weight matrix $W_i \in \mathbb{R}^{d_{i+1} \times d_i}$, bias $b_i \in \mathbb{R}^{d_{i+1}}$, and activation function $\sigma_i : \mathbb{R}^{d_{i+1}} \to \mathbb{R}^{d_{i+1}}$, where $d_i \in \mathbb{N}, d_1 = d, d_n = m$. Define the hidden layers as

$$h_k(x) = \sigma_k(W_k x + b_k),$$

for all $k \in [n]$. Clearly, $f = h_n \circ h_{n-1} \circ \ldots \circ h_1$. And our intermediate spaces are simply $\{\mathbb{R}^{d_i}\}_{i=1}^n$. It remains to define a metric on these hidden spaces. There are many ways of doing this. For example,

- For any $1 \le p \le \infty$, endow each intermediate space with the $\ell_p$-norm.
- Define $d(x, y) = 1 - \cos(\theta_{x,y})$ where $\theta_{x,y}$ is the angle between $x, y$. Then, if we choose $\sigma_i$ to restrict the image of $h_i$ to be the unit sphere, we may endow each intermediate space with this *cosine distance*.

Note here that there are two steps here: we first identify what the intermediate spaces are, then assign metrics to them. The process of identfying these intermediate spaces may be independent of the metrics we end of assigning them.

**Convolutional Neural Network.** For simplicity, we only consider the case of a single 2d-convolution layer, a convolutional network with higher dimensions or more layers can be derived inductively. Let $f : \mathbb{R}^{c \times h \times w} \to \mathbb{R}^{c' \times h' \times w'}$. Suppose that this layer is parameterized by kernels $W_i \in \mathbb{R}^{k \times k}$ for $i \in [c']$ and some $k \in \mathbb{N}$ as well as a bias $b \in \mathbb{R}^{c'}$. Then, it follows that

$$f(x)_j = \left( \mathbf{1}_{h' \times w'} b_j + \sum_{i=1}^c W_j * x[i, :, :] \right),$$

for $j \in [c']$ where $f(x)_j \in \mathbb{R}^{h' \times w'}$ for $h', w'$ being the resulting dimension after convolution with a $k \times k$ kernel. Here, $\mathbf{1}_{h' \times w'} \in \mathbb{R}^{h' \times w'}$ is a one matrix. To induce a distance metric on this output space, we can simply define a matrix norm on each of the output channels and sum them. Let $\{\|\cdot\|_i\}_{i=1}^{c'}$ be a collection of matrix norms. Then, we define

$$d(f(x), f(x')) = \sum_{i=1}^{c'} \|f(x)_i - f(x')_i\|_i.$$

It is easy to verify that this is a metric. Thus, the availability of a metric decomposition is not affected by parameter sharing.

Instead of incorporating every individual channel into our metric, we may also consider applying a pooling operation before passing the result through a single matrix norm, $\|\cdot\|$. For example,

$$d(f(x), f(x')) = \frac{1}{c'} \left\| \sum_{i=1}^{c'} f(x)_i - \sum_{i=1}^{c'} f(x')_i \right\|.$$

This, however, is no longer a metric, as definiteness is not preserved. That is, there exists $f(x) \ne f(x')$ where $d(f(x), f(x')) = 0$. This issue can be easily resolved by having $d(\cdot, \cdot)$ operate on a quotient space with respect to the equivalence relation $f(x) \sim f(x')$ if and only if $\sum_{i=1}^{c'} f(x)_i = \sum_{i=1}^{c'} f(x')_i$. This technique is further explored in the next subsection.

**Residual Connections.** We present two distinct metric decompositions of a residual network. Consider two fully-connected layers with one residual connection. This is visualized below.

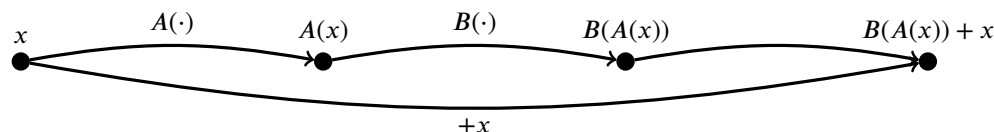

Let us assume that $A : \mathbb{R}^{d_1} \to \mathbb{R}^{d_2}$ and $B : \mathbb{R}^{d_2} \to \mathbb{R}^{d_1}$. Here, the input $x$ feeds back into the output layer $B$ creating a residual block (the set of layers between the input and the residual connection).

Trivially, we can aggregate the entire residual block as one metric decomposition. That is, let $h(x) = B(A(x)) + x$ be our metric decomposition. Then, define a metric on the image of $h$, $\mathbb{R}^{d_1}$, analogous to the hidden layers of a fully-connected neural network. This is the approach we use throughout our practical applications section (Section 5), and it is the standard way to counter layers in computer vision [27] and natural langauge processing [16].

To operate at a finer lever of granularity, we can also represent each layer within the residual block as a part of a metric decomposition. Let us redefine the residual block such that at every layer, we keep track of the input. The computational graph for this is shown below.

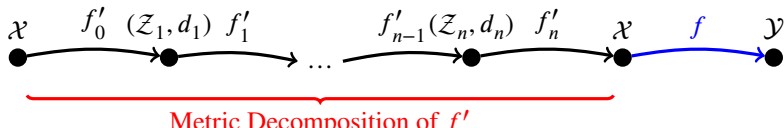

Define $A' : x \mapsto (A(x), x)$, $B' : (A(x), x) \mapsto (B(A(x)), x)$ and $x' : (B(A(x)), x) \mapsto (B(A(x))+x, x)$. Then, it follows that $x \to A' \to B' \to x'$ forms a metric decomposition. Here, the metric in each layer is with respect to the quotient space where $(a, a') \sim (b, b')$ if and only if $a = b$. Therefore, we also recover the same vector space structure.

**Transformers.** By chaining our metric decompositions for the residual blocks with our metric decompositions for the fully-connected networks we can easily create a metric decomposition for any transformer. Throughout the paper, we use two distinct methods to generate representations of its hidden layers:

- After each attention block which consists of multiheaded attention and multilayered percep-trons, we retrieve the last token.
- We average all of the tokens together.

In both of these methods, we are significantly reducing the dimension of the hidden layer. Thus, to formalize these metrics, we need to quotient out points that break the definiteness of our metric, as we have done before with the residual block.

## A.2 Beyond Neural Networks: Inducing Metric Decompositions

We have shown that our notion of a metric decomposition can well-describe many deep learning architectures, but what about models that are not neural networks (like a decision tree)? Herein, we demonstrate that we can induce metric decompositions even when the model itself does not have explicit hidden layers.

Let us now consider an arbitrary function $f : \mathcal{X} \to \mathcal{Y}$. We can induce a metric decomposition on $f$ through an auxiliary function $f' : \mathcal{X} \to \mathcal{X}$, for a metric-decomposable $f'$. If $f' = \text{Id}_{\mathcal{X}}$, then, $f = f \circ f'$ and the metric decomposition of $f$ would be exactly the metric decomposition of $f'$. This is visualized below.

$$\mathcal{X} \xrightarrow{f'_0} (\mathcal{Z}_1, d_1) \xrightarrow{f'_1} \cdots \xrightarrow{f'_{n-1}} (\mathcal{Z}_n, d_n) \xrightarrow{f'_n} \mathcal{X} \xrightarrow{f} \mathcal{Y}$$

$$\underbrace{\qquad\qquad\qquad\qquad\qquad}_{\text{Metric Decomposition of } f'}$$

Essentially, we have created an autoencoder for $\mathcal{X}$. This is common in many applications where a neural network or some other method is used as a feature extractor. In this way, we can simply define our metric with respect to these extracted features. However, this requires that either the autoencoder

to be exact or that our function $f$ is invariant under representations that collide. Thus, this would allow models such as decision trees to also be metric decomposed.

# B  Proof of Robustness

**Theorem** (See Thm. 4.1). *Let $A \subset \mathcal{X} \times \mathcal{Y}$ such that $\mathbb{P}_{D_{\mathcal{X},\mathcal{Y}}}[A] > 0$ and $\delta, \eta > 0$. Let $A' = \{(x', y') \in \mathcal{X} \times \mathcal{Y} : \mathbb{E}_{\substack{(x,y) \sim D_{\mathcal{X},\mathcal{Y}} \\ (x,y) \in A}} \Delta \mathcal{L}_f^{(x,y)}(x', y') > \eta\}$. If $f : \mathcal{X} \to \mathcal{Y}$ is $\epsilon$-knowledge continuous with respect to the hidden layer indexed by $k$ and $(\mathcal{Z}_k, d_k)$ is bounded by $B > 0$, then*

$$\mathbb{P}_{(x,y) \sim D_{\mathcal{X},\mathcal{Y}}}[A' \mid d_k(f^k(x), f^k(A)) < \delta] \leq \frac{\epsilon \delta}{\eta \left(1 - \exp\left[-\Omega\left(\frac{\delta}{B} - \sqrt{\log \frac{1}{\mathbb{P}[A]}}\right)^2\right]\right)}, \tag{B.1}$$

*where $f^k(A) = \{f^k(a) : a \in A\}$.*

*Proof.*

$$\mathbb{P}_{(x,y) \sim D_{\mathcal{X},\mathcal{Y}}}\left[A' \mid d_k(f^k(x), f^k(A)) < \delta\right] = \frac{\mathbb{P}_{(x,y) \sim D_{\mathcal{X},\mathcal{Y}}}\left[A' \cap d_k(f^k(x), f^k(A)) < \delta\right]}{\mathbb{P}_{(x,y) \sim D_{\mathcal{X},\mathcal{Y}}}[d_k(f^k(x), f^k(A)) < \delta]}. \tag{B.2}$$

We bound the numerator and denominator of Eq. B.2 separately. The denominator is given by Cor. B.3. We upper-bound the numerator using Markov's inequality. Firstly, we find the expectation of $\mathcal{L}_f^{(x,y)}(x', y')$ over $A' \cap d_k(f^k(x), f^k(A)) < \delta$:

$$\mathbb{E}_{(x,y) \sim D_{\mathcal{X},\mathcal{Y}}} \sigma_f^k(x, y) = \mathbb{E}_{(x,y) \sim D_{\mathcal{X},\mathcal{Y}}}\left(\mathbb{E}_{(x',y') \sim D_{\mathcal{X},\mathcal{Y}}}\left[\frac{\Delta \mathcal{L}_f^{(x,y)}(x', y')}{d_k(f^k(x), f^k(x'))}\right]\right), \tag{B.3}$$

$$= \mathbb{E}_{(x,y),(x',y') \sim (D_{\mathcal{X},\mathcal{Y}} \times D_{\mathcal{X},\mathcal{Y}})}\left[\frac{\Delta \mathcal{L}_f^{(x,y)}(x', y')}{d_k(f^k(x), f^k(x'))}\right]. \tag{B.4}$$

The previous inequality follows from Fubini's theorem, then

$$\mathbb{E}_{(x,y) \sim D_{\mathcal{X},\mathcal{Y}}} \sigma_f^k(x, y) \geq \mathbb{E}_{\substack{(x,y),(x',y') \sim (D_{\mathcal{X},\mathcal{Y}} \times D_{\mathcal{X},\mathcal{Y}}) \\ (x',y') \in A \\ d_k(f^k(x), f^k(A)) < \delta}}\left[\frac{\Delta \mathcal{L}_f^{(x,y)}(x', y')}{d_k(f^k(x), f^k(x'))}\right], \tag{B.5}$$

$$\geq \frac{1}{\delta} \mathbb{E}_{\substack{(x,y),(x',y') \sim (D_{\mathcal{X},\mathcal{Y}} \times D_{\mathcal{X},\mathcal{Y}}) \\ (x',y') \in A \\ d_k(f^k(x), f^k(A)) < \delta}}\left[\Delta \mathcal{L}_f^{(x,y)}(x', y')\right], \tag{B.6}$$

$$\delta \, \mathbb{E}_{(x,y) \sim D_{\mathcal{X},\mathcal{Y}}} \sigma_f^k(x, y) \geq \mathbb{E}_{\substack{(x,y),(x',y') \sim (D_{\mathcal{X},\mathcal{Y}} \times D_{\mathcal{X},\mathcal{Y}}) \\ (x',y') \in A \\ d_k(f^k(x), f^k(A)) < \delta}}\left[\Delta \mathcal{L}_f^{(x,y)}(x', y')\right]. \tag{B.7}$$

And by $\epsilon$-knowledge continuity,

$$\delta \epsilon \geq \mathbb{E}_{\substack{(x,y),(x',y') \sim (D_{\mathcal{X},\mathcal{Y}} \times D_{\mathcal{X},\mathcal{Y}}) \\ (x',y') \in A \\ d_k(f^k(x), f^k(A)) < \delta}}\left[\Delta \mathcal{L}_f^{(x,y)}(x', y')\right]. \tag{B.8}$$

This gives us an upper-bound of expectation of $\Delta \mathcal{L}_f^{(x,y)}(x', y')$ over the set of all points that are within $\delta$-radius from $f^k(A)$. Since $\Delta \mathcal{L}_f^{(x,y)}(x', y') \geq 0$ everywhere, by Markov's inequality,

$$\mathbb{P}_{(x,y) \sim D_{\mathcal{X},\mathcal{Y}}}[A' \cap d_k(f^k(x), f^k(A)) < \delta] \leq \frac{\delta \, \mathbb{E} \, \sigma_f^k(x, y)}{\eta}, \tag{B.9}$$

$$\leq \frac{\delta \epsilon}{\eta}. \tag{B.10}$$

The last inequality follows from $\mathbb{E}_{(x,y)\sim D_{\mathcal{X},\mathcal{Y}}} \sigma_f^k(x, y) < \epsilon$, by the definition of $\epsilon$-knowledge continuity. Now, by applying the complement of Lem. B.2, we lower-bound the denominator and yield the following

$$\mathbb{P}_{(x',y')\sim D} \left[ A' \mid d_k(f^k(x), f^k(x')) < \delta \right] \leq \frac{\epsilon \delta}{\eta \left( 1 - \exp\left( -\frac{2}{B^2} \left( \delta - B\sqrt{\frac{1}{2} \log \frac{2}{\mathbb{P}[A]}} \right)^2 \right) \right)}. \tag{B.11}$$

The proof is concluded by applying $\Omega(\cdot)$ notation to the denominator. ■

## B.1 Technical Lemmas

**Definition 6.** *A function* $f : \mathcal{X}_1 \times \dots \times \mathcal{X}_n \to \mathbb{R}$ *has bounded variation if there are* $c_1, \dots, c_n \in \mathbb{R}$ *such that for all* $1 \leq i \leq n$ *and* $x_1 \in \mathcal{X}_1, \dots, x_n \in \mathcal{X}_n$,

$$\sup_{x_i' \in \mathcal{X}_i} |f(x_1, \dots, x_i, \dots, x_n) - f(x_1, \dots, x_i', \dots, x_n)| \leq c_i. \tag{B.12}$$

**Lemma B.1** (McDiarmid's Inequality). *Assume that the function* $f : \mathcal{X}_1 \times \dots \times \mathcal{X}_n \to \mathbb{R}$ *satisfy the bounded differences property with bounds* $c_1, \dots, c_n$. *Consider the independent random variables* $X_1, \dots, X_n$ *where* $X_i \in \mathcal{X}_i$ *for all* $1 \leq i \leq n$. *Then, for any* $\epsilon > 0$,

$$\mathbb{P}[|f(X_1, \dots, X_n) - \mathbb{E}[f(X_1, \dots, X_n)]| \geq \epsilon] \leq 2 \exp\left( -\frac{2\epsilon^2}{\sum_{i=1}^n c_i^2} \right). \tag{B.13}$$

**Lemma B.2.** *Suppose that* $(\mathcal{X}, d)$ *is a bounded metric space such that* $\sup_{x,x' \in \mathcal{X}} d(x, x') < B$ *for some* $B > 0$. *Let* $A \subset X$ *such that* $\mathbb{P}[A] > 0$ *and* $\epsilon > 0$. *Then,*

$$\mathbb{P}[d(x, A) \geq \epsilon] \leq \exp\left( -\frac{2}{B^2} \left( \epsilon - B\sqrt{\frac{1}{2} \log \frac{2}{\mathbb{P}[A]}} \right)^2 \right).$$

*Proof.* For brevity, denote $f_A(x) = d(A, x) = \inf_{a \in A} d(x, a)$. Since $(\mathcal{X}, d)$ is a bounded metric space, by Lem. B.1,

$$\mathbb{P}[|f_A(x) - \mathbb{E}f_A(x)| \geq \epsilon] = 2 \exp\left( -\frac{2\epsilon^2}{B^2} \right), \tag{B.14}$$

$$\mathbb{P}[f_A(x) - \mathbb{E}f_A(x) \geq \epsilon] + \mathbb{P}[f_A(x) - \mathbb{E}f_A(x) \leq -\epsilon] \leq 2 \exp\left( -\frac{2\epsilon^2}{B^2} \right), \tag{B.15}$$

$$\mathbb{P}[f_A(x) - \mathbb{E}f_A(x) \leq -\epsilon] \leq 2 \exp\left( -\frac{2\epsilon^2}{B^2} \right), \tag{B.16}$$

Let $\epsilon = \mathbb{E}f_A(x)$. Then,

$$\mathbb{P}[f_A(x) \leq 0] \leq 2 \exp\left( -\frac{2(\mathbb{E}f_A(x))^2}{B^2} \right), \tag{B.17}$$

$$\mathbb{P}[A] \leq 2 \exp\left( -\frac{2(\mathbb{E}f_A(x))^2}{B^2} \right), \tag{B.18}$$

$$\mathbb{E}f_A(x) \leq \sqrt{\frac{B^2}{2} \log\left( \frac{2}{\mathbb{P}[A]} \right)}. \tag{B.19}$$

The second inequality follows from $\mathbb{P}[f_A(x) \leq 0] = \mathbb{P}[f_A(x) = 0] \geq \mathbb{P}[A]$. By Eq. B.15,

$$\mathbb{P}[f_A(x) - \mathbb{E}f_A(x) \geq \epsilon] + \mathbb{P}[f_A(x) - \mathbb{E}f_A(x) \leq -\epsilon] \leq 2 \exp\left( -\frac{2\epsilon^2}{B^2} \right),$$

$$\mathbb{P}[f_A(x) - \mathbb{E}f_A(x) \geq \epsilon] \leq 2\exp\left(-\frac{2\epsilon^2}{B^2}\right),$$

$$\mathbb{P}[f_A(x) \geq \epsilon + \mathbb{E}f_A(x)] \leq 2\exp\left(-\frac{2\epsilon^2}{B^2}\right),$$

$$\mathbb{P}\left[f_A(x) \geq \epsilon + \sqrt{\frac{B^2}{2}\log\left(\frac{2}{\mathbb{P}[A]}\right)}\right] \leq 2\exp\left(-\frac{2\epsilon^2}{B^2}\right), \qquad \text{(by Eq. B.19,)}$$

for any $\delta > 0$, let $\epsilon = \delta - \sqrt{\frac{B^2}{2}\log\frac{2}{\mathbb{P}[A]}}$. And so,

$$\mathbb{P}[f_A(x) \geq \delta] \leq 2\exp\left(-\frac{2}{B^2}\left(\delta - B\sqrt{\frac{1}{2}\log\left(\frac{2}{\mathbb{P}[A]}\right)}\right)^2\right),$$

which is the desired expression. $\blacksquare$

**Corollary B.3.** $\mathbb{P}[f_A(x) < \delta] \geq 1 - 2\exp\left(-\frac{2}{B^2}\left(\delta - B\sqrt{\frac{1}{2}\log\left(\frac{2}{\mathbb{P}[A]}\right)}\right)^2\right).$

## C  Proof of Expressiveness

**Proposition** (See Prop. 4.4). *Suppose* $(\mathcal{X}, d_{\mathcal{X}}), (\mathcal{Y}, d_{\mathcal{Y}}) := (\mathcal{X}, d_{\mathcal{X}})$ *are* **compact** *metric spaces,* $\mathcal{F} \subset \mathcal{Y}^{\mathcal{X}}$ *is the* **set of all continuous functions** *from* $\mathcal{X}$ *to* $\mathcal{Y}$ *such that* $\int d_{\mathcal{X}}(x, x')^{-1} d\mu_f < \infty$ *and* $\mathcal{L}$ *be Lipschitz continuous in both coordinates. Then, there exists a universal function approximator* $\mathcal{U}$ *of* $\mathcal{F}$ *that is knowledge continuous (i.e.* $\mathbb{E}\,\sigma_f^k(x, y) < \infty$ *for some $k$).*

*Proof.* By Lem. C.3, the set of Lipschitz continuous functions $\mathscr{L}$ is dense in the set of all continuous functions $\mathscr{C}$ with respect to the uniform metric. By Lem. C.1, since $|\mathcal{L}(x, y)| \leq K d(x, y)$, if $\sup_{x \in \mathcal{X}} d(f(x), g(x)) < \epsilon$, then for any probability measure $\mathbb{P}$ over $\mathcal{X}$,

$$\int \mathcal{L}(f(x), g(x))\, d\mathbb{P} \leq \int |\mathcal{L}(f(x), g(x))|\, d\mathbb{P} \leq K\epsilon,$$

where $K$ is the Lipschitz constant of $\mathcal{L}$. This implies that for any sequence $\epsilon_1, \epsilon_2, \ldots$ we can choose Lipschitz continuous functions $f_1, f_2, \ldots$ with Lipschitz constants $C_1, C_2, \ldots$ such that $\int \mathcal{L}(f_n(x), y)\, d\mu_f < \epsilon_n$. It remains to show that each of these functions are in fact knowledge continuous. Since $\mathcal{X}$ is a metric space, we consider the trivial metric decomposition of our sequence of functions (see Remark 1). Specifically, we denote $h_1 = \mathrm{Id}_{\mathcal{X}}$ and proceed to bound $\mathbb{E}\,\sigma_f^1(x, y)$.

$$\mathbb{E}\,\sigma_{f_n}^1(x, y) = \iint \frac{\Delta\mathcal{L}_{f_n}^{(x,y)}(x', y')}{d_{\mathcal{X}}(x, x')}\,(d\mu_f \times d\mu_f), \tag{C.1}$$

$$\leq \iint \frac{|\mathcal{L}(f_n(x), y) - \mathcal{L}(f_n(x'), y) + \mathcal{L}(f_n(x'), y) - \mathcal{L}(f_n(x'), y')|}{d_{\mathcal{X}}(x, x')}\,(d\mu_f \times d\mu_f), \tag{C.2}$$

$$\leq \iint \frac{|\mathcal{L}(f_n(x), y) - \mathcal{L}(f_n(x'), y)|}{d_{\mathcal{X}}(x, x')}\,d(\mu_f \times \mu_f) \tag{C.3}$$

$$+ \iint \frac{|\mathcal{L}(f_n(x'), y) - \mathcal{L}(f_n(x'), y')|}{d(x, x')}\,(d\mu_f \times d\mu_f), \tag{C.4}$$

$$\leq \iint \frac{K d_{\mathcal{X}}(f(x), f(x'))}{d_{\mathcal{X}}(x, x')}\,d(\mu_f \times \mu_f) + \iint \frac{K d_{\mathcal{X}}(y, y')}{d_{\mathcal{X}}(x, x')}\,d(\mu_f \times \mu_f), \tag{C.5}$$

By Lem. C.4, any compact metric space is bounded. So, let $(\mathcal{X}, d)$ be bounded by $b > 0$. It follows that $d_{\mathcal{X}}(y, y') \leq b$ and

$$\leq \iint K C_n\, d(\mu_f \times \mu_f) + K b \int \frac{1}{d_{\mathcal{X}}(x, x')}\,d\mu_f, \tag{C.6}$$

$$= KC_n + Kb \int d_{\mathcal{X}}(x, x')^{-1} d\mu_f, \tag{C.7}$$

By assumption $\int d_{\mathcal{X}}(x, x')^{-1} d\mu_f < \infty$ and the statement of the proposition follows. ∎

## C.1 Technical Lemmas

**Lemma C.1.** *If $\mathcal{L}(\cdot, \cdot)$ is Lipschitz continuous in both coordinates, then for any $x, x' \in \mathcal{X}$, $|\mathcal{L}(x, x')| \leq K d(x, x')$, where $K$ is the Lipschitz constant of $\mathcal{L}$.*

*Proof.* By Lipschitz continuity,

$$|\mathcal{L}(x, x') - \mathcal{L}(x, x)| \leq K d(x, x'),$$
$$|\mathcal{L}(x, x')| \leq K d(x, x').$$

∎

**Lemma C.2.** *The set of all Lipschitz continuous functions from $\mathcal{X} \to \mathcal{X}$ separates all points in $\mathcal{X}$.*

*Proof.* The identity function is 1-Lipschitz continuous and it also separates all points in $\mathcal{X}$. ∎

**Corollary C.3.** *Let $\mathscr{C} \subset \mathcal{X}^{\mathcal{X}}$ be the set of all continuous functions from $\mathcal{X} \to \mathcal{X}$ and $\mathscr{L} \subset \mathcal{X}^{\mathcal{X}}$ be the set of all Lipschitz continuous functions from $\mathcal{X} \to \mathcal{X}$. If $\mathcal{X}$ is compact, then $\mathscr{L}$ is dense in $\mathscr{C}$ with respect to the uniform metric: $d'(f, g) = \sup_{x \in \mathcal{X}} d(f(x), g(x))$.*

*Proof.* This follows directly from Lem. C.2 and the Stone-Weierstrass theorem [65]. ∎

**Lemma C.4.** *Any compact metric space $(\mathcal{X}, d)$ is also bounded.*

*Proof.* By way of contraposition suppose that $(\mathcal{X}, d)$ is not bounded. Then, $\sup_{x, x' \in \mathcal{X}} d(x, x') = \infty$. Pick $x_1 \in \mathcal{X}$ arbitrarily and pick $x_n$ for $n \in \mathbb{Z}^+, n > 1$ such that $d(x_n, x_1) > n$. Clearly, there does not exist a convergent subsequence of the sequence $x_1, x_2, \dots$. Thus, $(\mathcal{X}, d)$ cannot be compact. ∎

# D Proof of Equivalence Between Lipschitz Continuity and Knowledge Continuity

**Proposition.** *(See Prop. 4.6) Suppose that $(\mathcal{X}, d_{\mathcal{X}}), (\mathcal{Y}, d_{\mathcal{Y}})$ are metric spaces. Let the first $n$ metric decompositions of $f : \mathcal{X} \to \mathcal{Y}$ be $K_i$-Lipschitz continuous, for $i \in [n]$. If $f$ is $\epsilon$-knowledge continuous with respect to the $n^{th}$ hidden layer and $d_{\mathcal{Y}}(f(x), f(x')) \leq \eta \Delta \mathcal{L}_f^{(x,y)}(x', y)$ for all $x, x' \in \mathcal{X}, y \in \mathcal{Y}$, and some $\eta > 0$, then $f$ is Lipschitz continuous in expectation. That is,*

$$\mathbb{E}_{(x,y),(x',y') \sim \mathcal{D}_{\mathcal{X},\mathcal{Y}}} \frac{d_{\mathcal{Y}}(f(x), f(x'))}{d_{\mathcal{X}}(x, x')} \leq \epsilon \eta \prod_{j=1}^{n} K_j. \tag{D.1}$$

*Proof.* We proceed to bound the knowledge continuity of $f$ from below.

$$\mathbb{E} \, \sigma_f^k(x, y) \geq \mathbb{E}_{(x,y) \sim \mathcal{D}_{\mathcal{X},\mathcal{Y}}} \mathbb{E}_{\substack{(x',y') \sim \mathcal{D}_{\mathcal{X},\mathcal{Y}} \\ y'=y}} \frac{\Delta \mathcal{L}_f^{(x,y)}(x', y)}{d_k(f^k(x), f^k(x'))}, \tag{D.2}$$

$$\geq \mathbb{E}_{(x,y) \sim \mathcal{D}_{\mathcal{X},\mathcal{Y}}} \mathbb{E}_{\substack{(x',y') \sim \mathcal{D} \\ y'=y}} \frac{\Delta \mathcal{L}_f^{(x,y)}(x', y)}{\prod_{j=1}^{n} K_j d_{\mathcal{X}}(x', x)}, \tag{D.3}$$

$$\geq \mathbb{E}_{(x,y) \sim \mathcal{D}_{\mathcal{X},\mathcal{Y}}} \mathbb{E}_{\substack{(x',y') \sim \mathcal{D} \\ y'=y}} \frac{\frac{1}{\eta} d_{\mathcal{Y}}(f(x), f(x'))}{\prod_{j=1}^{n} K_j d_{\mathcal{X}}(x, x')}, \tag{D.4}$$

$$= \mathbb{E}_{(x,y),(x',y') \sim \mathcal{D}_{\mathcal{X},\mathcal{Y}}} \frac{\frac{1}{\eta} d_{\mathcal{Y}}(f(x), f(x'))}{\prod_{j=1}^{n} K_j d_{\mathcal{X}}(x, x')}. \tag{D.5}$$

Eq. D.2 comes from the fact that we take the expectation only over pairs of points $(x, y), (x', y')$ where $y = y'$ and also because the summand is always nonnegative. Then, we inductively apply the definition of $K_i$-Lipschitz continuity to yield Eq. D.3. Eq. D.4 follows directly from the assumption in the statement of the proposition. Since the expression in Eq. D.4 now has no dependence on the label distribution, we may expand the expectation which results in Eq. D.5. Lastly, by the definition of $\epsilon$-knowledge continuity,

$$\epsilon \geq \mathbb{E}_{(x,y),(x',y') \sim \mathcal{D}_{\mathcal{X},\mathcal{Y}}} \frac{\frac{1}{\eta} d_{\mathcal{Y}}(f(x), f(x'))}{\prod_{j=1}^{n} K_j d_{\mathcal{X}}(x, x')},$$

$$\epsilon \eta \prod_{j=1}^{n} K_j \geq \mathbb{E}_{(x,y),(x',y') \sim \mathcal{D}_{\mathcal{X},\mathcal{Y}}} \frac{d_{\mathcal{Y}}(f(x), f(x'))}{d_{\mathcal{X}}(x, x')},$$

and this concludes the proof of the proposition. ∎

To prove Cor. 4.7, we need the following auxiliary result from [89].

**Proposition D.1** (See [89]). *For a neural network $f : \mathbb{R}^n \to \mathbb{R}^K$ with Lipschitz constant $L$ under $\ell_p$-norm, define the resulting classifier $g$ as $g(x) := \arg\max_{k \in [K]} f_k(x)$ for an input $x$. Then, $g$ is provably robust under perturbations $\|\delta\|_p < \frac{\sqrt[p]{2}}{2L} margin(f(x))$, i.e.*

$$g(x + \delta) = g(x) \qquad for\ all\ \|\delta\|_p < \frac{\sqrt[p]{2}}{2L} margin(f(x)). \tag{D.6}$$

*Here, $margin(f(x))$ is the difference between the largest and second largeset output logit.*

**Corollary** (See Cor. 4.7). *Suppose that assumptions of Prop. 4.6 are true. And also assume that $(\mathcal{X}, d_{\mathcal{X}}) = (\mathbb{R}^n, \ell_p)$, $(\mathcal{Y}, d_{\mathcal{Y}}) = (\mathbb{R}^m, \ell_p)$, for $1 \leq p \leq \infty$. Define a classifier from $f : \mathbb{R}^n \to \mathbb{R}^m$, $g$, where $g(x) := \arg\max_{k \in [m]} f_k(x)$ for any $x \in \mathbb{R}^n$. Then, with probability $1 - \frac{\epsilon \eta}{t} \prod_{j=1}^{n} K_j$, $g(x) = g(x + \delta)$ for all $\|\delta\|_p < \frac{\sqrt[p]{2}}{2t} margin(f(x))$ and $t > 0$. $f_k(x)$ is the $k^{th}$ coordinate of $f(x)$ and $margin(f(x))$ denotes the difference between the largest and second-largest output logits.*

*Proof.* By Prop. 4.6, we have that

$$\mathbb{E}_{(x,y),(x',y') \sim \mathcal{D}_{\mathcal{X},\mathcal{Y}}} \frac{d_{\mathcal{Y}}(f(x), f(x'))}{d_{\mathcal{X}}(x, x')} \leq \epsilon \eta \prod_{j=1}^{n} K_j. \tag{D.7}$$

By Markov's inequality,

$$\mathbb{P}_{(x,y),(x',y') \sim \mathcal{D}_{\mathcal{X},\mathcal{Y}}} \left[ \frac{d_{\mathcal{Y}}(f(x), f(x'))}{d_{\mathcal{X}}(x, x')} \geq t \right] \leq \frac{\epsilon \eta}{t} \prod_{j=1}^{n} K_j. \tag{D.8}$$

We yield the corollary by directly applying Prop. D.1 assuming that $f$ is $t$-Lipschitz continuous. ∎

Next, we establish conditions under which Lipschitz continuity implies knowledge continuity.

**Proposition** (Prop. 4.8). *Let $(\mathcal{X}, d_{\mathcal{X}}), (\mathcal{Y}, d_{\mathcal{Y}})$ be a metric spaces. Let $f : \mathcal{X} \to \mathcal{Y}$ be $\epsilon$-Lipschitz continuous and $\mathcal{L}(f(x), y)$ be $\eta$-Lipschitz continuous with respect to both coordinates. If the first $n$ metric decompositions of $f$ are $K_i$-Lipschitz continuous, then $f$ is knowledge continuous with respect to the $n^{th}$ hidden layer. That is,*

$$\mathbb{E}_{(x,y) \sim \mathcal{D}_{\mathcal{X},\mathcal{Y}}} \sigma_f^n(x, y) \leq \epsilon \eta \prod_{j=1}^{n} \frac{1}{K_j}. \tag{D.9}$$

*Proof.* Let us start with the definition of $\epsilon$-Lipschitz continuity and lower-bound it. For any $(x, y), (x', y') \in \mathcal{X} \times \mathcal{Y}$,

$$\frac{d_{\mathcal{Y}}(f(x), f(x'))}{d_{\mathcal{X}}(x, x')} \leq \epsilon, \tag{D.10}$$

$$\frac{d_{\mathcal{Y}}(f(x), f(x'))}{\prod_{j=1}^{n} \frac{1}{K_j} d_k(f^k(x), f^k(x'))} \leq \epsilon, \tag{D.11}$$

$$\frac{\frac{1}{\eta} |\mathcal{L}(x, y) - \mathcal{L}(x', y')|}{\prod_{j=1}^{n} \frac{1}{K_j} d_k(f^k(x), f^k(x'))} \leq \epsilon, \tag{D.12}$$

$$\frac{|\mathcal{L}(x, y) - \mathcal{L}(x', y')|}{d_k(f^k(x), f^k(x'))} \leq \epsilon \eta \prod_{j=1}^{n} \frac{1}{K_j}. \tag{D.13}$$

Eq. D.11 follows from inductively applying the definition of Lipschitz continuity on the metric decompositions of $f$. Specifically, $d_{i+1}(f^{i+1}(x), f^{i+1}(x')) \leq K_i d_i(f^i(x), f^i(x))$. Then, by the Lipschitz continuity of $\mathcal{L}$ in both coordinates we yield Eq. D.12. Since the Lebesgue integral preserves order, Eq. D.13 directly implies the statement of the proposition and this concludes the proof. ∎

## E   Predicting Adversarial Robustness with Volatility

In this section, we detail the experimental methods and results that use knowledge continuity to predict adversarial vulnerability, briefly discussed in Section 5. We focus on langauge models of various sizes and their ability to perform sentiment classification on the IMDB dataset [48]. Before computing any statistics of the model, we finetune it against the IMDB dataset and reserve a test set on which we compute a *vulnerability score* and estimate the model's adversarial vulnerability.

**Vulnerability Score.** As described in the main text, given a model with $n$ hidden layers, we compute all of its $k$-volatility scores. This is done with a naive Monto-Carlo algorithm which we present in Appendix G. This results in a list of $k$-volatility scores $\{\epsilon_1, \ldots, \epsilon_n\}$, one for each hidden layer. Then, we perform a simple average $n^{-1} \sum_{k=1}^{n} \epsilon_k$. Let us denote this quantity as the *vulnarability score*.

**Estimating Adversarial Robustness.** It remains to estimate the adversarial vulnerability of a given model. We do this empirically by applying an out-of-the-box adversarial attack (specifically, TextFooler [35]) on the given model with respect to the reserved test set. We then measure the number of successful adversarial attacks defined as

$$\sharp \text{Successful Adversarial Attacks} = \frac{|\mathcal{X}^{\text{adversarial}} \cap \mathcal{X}^{\text{correct}})}{|\mathcal{X}^{\text{correct}}|},$$

where $\mathcal{X}^{\text{correct}}$ is the set of examples in the test set that are correctly classified by the model (after finetuning) without any intervention. And, $\mathcal{X}^{\text{adversarial}}$ are the set of examples that are incorrectly classified after an adversarial attack is applied. In other words, we only consider points where a perturbation will *worsen* performance. In expectation, this estimate of adversarial robustness should be a $1/2$ factor of the notion of vulnerability we present in Thm. 4.1, where we also consider a point to be vulnerable if perturbation *increases* its performance.

We then perform a linear regression using vulnerability score and a host of other model properties to predict the number of successful adversarial attacks. Concretely, we seek to learn the relationship:

$$\sharp \text{Successful Adversarial Attacks} = m^T \left( n^{-1} \sum_{k=1}^{n} \epsilon_k \oplus \underbrace{\ldots}_{\text{additional architectural variables}} \right) + b,$$

where $m \in \mathbb{R}^d$ and $b \in \mathbb{R}$ are the learnable regression parameters. We also incorporate $d - 1$ size and architectural variables into our regression as we found that significantly increases its predictiveness. And so, the input variables to our regression and their types are:

| Feature | Type |
|---|---|
| Encoder Only | $\{0, 1\}$ |
| Decoder Only | $\{0, 1\}$ |
| Encoder-Decoder | $\{0, 1\}$ |
| log($\sharp$Parameters) | $\mathbb{R}$ |
| $n^{-1} \sum_{k=1}^{n} \epsilon_k$ | $\mathbb{R}$ |

| Variables | (1) Coefficients | $\Delta R^2$ | (2) Coefficients | $\Delta R^2$ | (3) Coefficients | $\Delta R^2$ |
|---|---|---|---|---|---|---|
| `Encoder Only` | ✗ | ✗ | 1485 | 0.40 | −548 | 0.07 |
| `Decoder Only` | ✗ | ✗ | −2816 | 0.71 | −557 | 0.02 |
| `Encoder-Decoder` | ✗ | ✗ | 1332 | 0.29 | 1105 | 0.18 |
| `log(♯Parameters)` | ✗ | ✗ | 66 | $-6.1 \times 10^{-5}$ | −363 | 0.04 |
| $n^{-1} \sum_{k=1}^{n} \epsilon_k$ | **49** | **0.35** | 96 | 2.57 | ✗ | ✗ |
| $R^2$ | | 0.35 | | 0.48 | | 0.28 |

Table 2: Regression results from our three previously described experimental settings. We regress the number of successful adversarial attacks against (1) only the vulnerability score (2) vulnerability score and model characteristics (3) only model characteristics. The coefficients for each of these regressions results are shown in the column *Coefficients*. We also run permutation tests for each coefficient and the change in $R^2$ is shown in the column $\Delta R^2$ (higher the better).

For the regression itself, we perform a Ridge regression with $\alpha = 1$. We test three experimental conditions where we regress the model's adversarial robustness against: (1) only vulnerability score, (2) vulnerability score and model characteristics, (3) only model characteristics. We experiment with seven models: RoBERTa (Base/Large) [44], BERT-Uncased (Base/Large) [16], GPT2, and T5 (Small/Base) [58]. Our regression results are shown in Table. 2.

After yielding an initial line-of-best fit (see Fig. 2(a)), we run permutation tests to determine the contribution of each feature to the explained variance. Specifically, for each feature, keeping all else constant, we permute its values. If this feature is a significant contributor to the explained variance, intuitively, we should see a large decrease in $R^2$ after this intervention. If $s$ is the $R^2$ without any intervention and $s_{\sigma_i(d)}$ is the $R^2$ after permuting the data by $\sigma_i(\cdot) : [n] \to [n]$ (for a dataset of $n$ data points). Then, we define

$$\Delta R^2 = s - \frac{1}{N} \sum_{k=1}^{N} s_{\sigma_k(d)},$$

where $N$ controls the number of permutations that we apply. For all experiments we choose $N = 100$. For formal theory on permutation tests, see [8].

We find that when our *vulnerability score* is added to the regression, it contributes significantly to the explained variance. Moreover, in (2), we see that vulnerability score has the highest feature importance among all regression variables.

## F    Localizing Volatile Hidden Representations

In this section, we localize adversarially vulnerable hidden representations in two ways. Firstly, we use $k$-volatility to gauge which layers are vulnerable across a selection of models. Then, we focus on model-specific characterizations of robustness with respect to $k$-volatility. We present experiments on the same selection of models in Appendix E, the same dataset (IMDB [48]), and the same adversarial attack (TextFooler [35]) to empirically measure adversarial vulnerability.

### F.1    Layerwise Volatility

As mentioned in the previous section (Section E), for a given model with $n$ hidden layers, we can measure its $k$-volatility for $k \in [n]$ through a Monte-Carlo algorithm. For each model, we then plot its $k$-volatility against its relative depth which is defined as $\lfloor k/n \rfloor$. These curves are shown in Fig. 2(b). We see that models which have different architectures independent of size have very different $k$-volatility curves.

We have already shown in the previous section that there is a positive correlation between $k$-volatility and adversarial vulnerability. However, this correlation is derived from the simple average of all $k$-volatility scores. Are the $k$-volatility scores in some layers more predictive of adversarial vulnerability

than others? If the $k$-volatility in some layers is more correlated with $k$-volatility in others, then it should suffice to minimize $k$-volatility in these former layers. This would also speed up regularization and training.

We repeat the experiments in the previous settings. But, instead of collating $k$-volatility through a simple average, we run one regression for each relative depth across all models (which we discretize into 9 bins). This result is shown in Fig. 2(c). Surprisingly, we find that the magnitude of $k$-volatility is not necessarily predictive of adversarial vulnerability. For example, in Fig. 2(b), almost all of the models exhibit low average $k$-volatility in the latter layers. However, the $k$-volatility of latter layers predict adversarial vulnerability the best.

### F.2 Model-Specific Volatility

We start by exploring the $k$-volatility across each of our test models. We notice that $k$-volatility cannot be predicted by surface-level features such as size or model type alone. This is shown clearly in Fig. 3. Yet, as discussed in Appendix E, it is still able to predict actual adversarial vulnerability with moderate power. Thus, we conjecture that $k$-volatility captures a complex aspect of the model's vulnerability which cannot be solely attributed to its size or type.

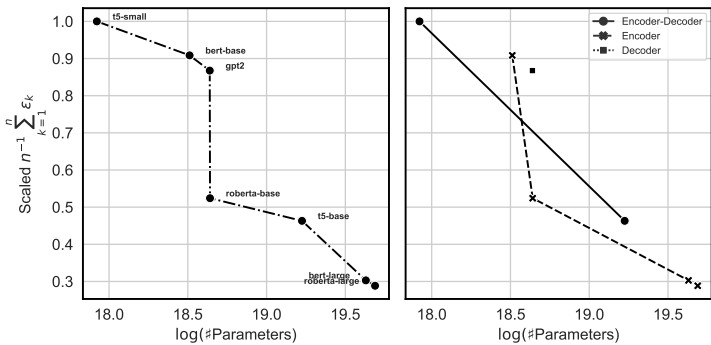

Figure 3: Average $k$-volatility plotted against the log of number of model parameters (left). We see that although is a strong negative correlation, the exactly relationship is nontrivial. Moreover, this negative correlation is also consistently observed across model families (right).

## G Regularizing Knowledge Continuity

In this section, we provide a comprehensive overview of regulating knowledge continuity to achieve robustness. We first show a simple algorithm that estimates $k$-volatility. Then, we demonstrate how this can incorporated into any loss function as a regularization term. We then prove guarantees that revolve around the unbiasedness of our estimation algorithm. Lastly, we present detailed discussion of the results shown in Table 1 including training details and ablation studies over the hyperparameters.

### G.1 Estimating Knowledge Continuity Algorithmically

We first present a method for estimating $k$-volatility. This is shown in Alg. 1(ESTKVOL). In theory, one should choose $M = N$, as this will lead to a most accurate estimate. This is similar to contrastive learning methods where it is desirable to make the minibatch sizes as large as possible [56]. However, if $N \gg 1$, this can become quickly intractable. In practice, during regularization we keep $N$ to be the same as if we were doing normal finetuning (i.e. 32/64) and set $M = N$. This works well, and, anecdotally, we find that in contrast to contrastive learning increasing $N$ or $M$ past this threshold yields marginal returns. Further work could examine this relationship in more detail.

As discussed in the main text, the choice of metric (or representation space) which we enforce knowledge continuity against is crucial as it determines the type of robustness we will achieve. Therefore, in Alg. 1(KCREG), we incorporate this detail by sampling a hidden layer of interest using a Beta distribution specified by hyperparameters $\alpha, \beta$. Then, on that minibatch, regularize $k$-volatility

**Algorithm 1** A Monte-Carlo algorithm for estimating $k$-volatility of some metric decomposable function $f$ with $n$ hidden layers (left). Augmenting any loss function to regularize $k$-volatility (right), given some Beta distribution parameterized by $\alpha, \beta$ and regularization strength $\lambda \geq 0$.

---

**procedure** ESTKVOL($\{(x_i, y_i)\}_{i=1}^N, M, f, k$)
    Sample $\{n_1, \ldots, n_M\} \subset [N]$ uniformly
    $\sigma_f^k \leftarrow 0$
    Losses $\leftarrow \{\mathcal{L}(f(x_{n_i}), y_{n_i})\}_{i=1}^M$
    **for** $(i, j) \in [M] \times [M]$ **do**
        Dist $\leftarrow d_k(f^k(x_{n_i}), f^k(x_{n_j}))$
        $\sigma_f^k \leftarrow \sigma_f^k + |\text{Losses}_i - \text{Losses}_j|/\text{DIST}$
    **return** $\sigma_f^k$

**procedure** KCREG($\alpha, \beta, M, \lambda$)
    $X \sim \text{Beta}(\alpha, \beta)$
    $k \leftarrow \max(\lfloor Xn \rfloor, 1)$
    $\sigma_f^k \leftarrow \text{ESTKVOL}(\{(x_i, y_i)\}_{i=1}^N, f, M, k)$
    **return** $\frac{1}{N} \sum_{i=1}^N \mathcal{L}(f(x_i), y_i) + \frac{1}{M^2} \lambda \sigma_f^k$

---

with respect to that sampled layer. Note that we choose the Beta distribution for simplicity, however, it can be replaced by any distribution like a mixture of Gaussians.

In contrast to existing adversarial training methods such as [32] and [63] which only use the embeddings, our algorithm gives the practitioner more control over which hidden layer (or distance metric) to enforce smoothness. In this way, if the practitioner has some knowledge *a priori* of the attacker's strategy, they may choose to optimize against the most suitable metric decomposition. We present a brief discussion of the various tradeoffs when choosing $\alpha, \beta$ in the following section as well as a detailed empirical analysis in the following subsections.

$\lambda$ is the weight we put on the regularizer in relation to the loss function $\mathcal{L}$. We provide a detailed ablation study of the effects of $\lambda$ in the following subsections. We surprisingly find that even for $\lambda \ll 1$ we can achieve significant edge in terms of robustness over existing methods. This is in contrast to virtual adversarial training methods such as [43] which requires applying a $\lambda$-value magnitudes larger. Moreover, for larger $\lambda$, we find that the accuracy of the model is not compromised. This provides some empirical support for Theorem 4.3.

### G.2 Theoretical Guarantees of $k$-Volatility Estimation

In this subsection, we show that our Monte-Carlo algorithm presented in Alg. 1(ESTKVOL) is an unbiased estimator. The proof is simple and follows from some bookkeeping.

**Proposition G.1** (Alg. 1(ESTKVOL) is an Unbiased Estimator). *Assuming that each data point in the batch, $\{(x_i, y_i)\}_{i=1}^N \sim \mathcal{D}_{\mathcal{X}, \mathcal{Y}}$, is sampled i.i.d., then Alg. 1(ESTKVOL) is an unbiased estimator for $\mathbb{E} \, \sigma_f^k(x, y)$.*

*Proof.* Let $\hat{\theta}$ be the random variable representing the output of Alg. 1. It suffices to show that

$$\mathbb{E}[\hat{\theta}] = \mathbb{E} \, \sigma_f^k(x, y),$$

where the expectation on the left-hand side is taken over the set of all batches. By the definition of Alg. 1(ESTKVOL),

$$\mathbb{E}[\hat{\theta}] = \mathbb{E}\left( \sum_{i=1}^M \sum_{j=1}^M \frac{1}{M^2} \frac{\Delta \mathcal{L}_f^{(x_{n_j}, y_{n_j})}(x_{n_i}, y_{n_i})}{d_k(f^k(x_{n_i}), f^k(x_{n_j}))} \right), \tag{G.1}$$

$$= \sum_{i=1}^M \sum_{j=1}^M \frac{1}{M^2} \mathbb{E}\left( \frac{\Delta \mathcal{L}_f^{(x_{n_j}, y_{n_j})}(x_{n_i}, y_{n_i})}{d_k(f^k(x_{n_i}), f^k(x_{n_j}))} \right), \tag{G.2}$$

$$= \mathbb{E} \, \sigma_f^k(x, y). \tag{G.3}$$

The second equality follows from the linearity of expectation. ∎

We emphasize that our estimator is very naive. Improving its efficiency could form the basis of possible future work. For example, Rao-Blackwellizing [6] Alg. 1(ESTKVOL) to yield an estimator with

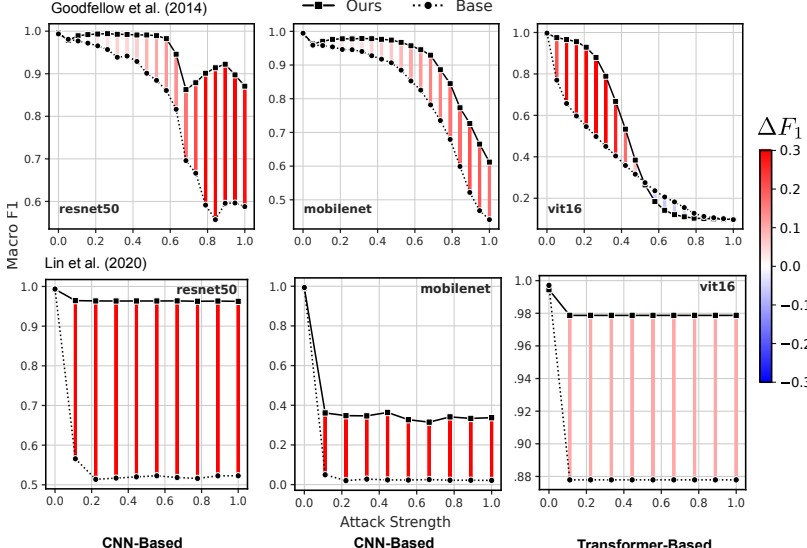

Figure 4: Regularization $k$-volatility for a host of vision models. We apply two adversarial attacks FGSM [24] (top row) and SI-NI-FGSM [41] (bottom row) with various attack strengths. Attack strength is measured in terms of maximum $\ell_2$-norm of the applied perturbation to the image.

smaller variance, applying rejection sampling to deal with the potential sparsity of the representation space discussed in Section 4.4, or adapting the regularization weight based on some bootstrapped confidence interval (if the estimate has higher variance then decrease weight on regularization and vice versa). However, we see that even with this naive algorithm we achieve improvements in robustness as well as training speed.

## G.3   Computer Vision Results

In addition to regulating language models, we also demonstrate that KCREG is effective for vision tasks. This provides empirical support for the equivalences we proved in Section 4.5. The exact same method of $k$-volatility estimation and loss augmentation is applied. We finetune three models ResNet50 [28], MobileNetV2 [61], and ViT16 [17] on the MNIST dataset both with and without our regularization algorithm. We then apply two different adversarial attacks: FGSM [24] and SI-NI-FGSM [41]. We find that in both cases, regularization $k$-volatility improves/stabilizes robustness across attack strengths (see Fig. 4).

## G.4   Ablation Studies

Herein, we present ablation studies for the crucial hyperparameters in our regularization algorithm (across the natural language tasks that we explored in the main text), Alg. 1(KCREG): $\lambda$ which is the weight we assign the knowledge continuity regulation loss and $(\alpha, \beta)$ which determines the sampling behavior of the index of the hidden representation space.

**Ablation Study of $\lambda$ (Fig. 5(right)).** The weight given to the regularizer ($\lambda$) is ablated over, with the results shown in Fig. 5. For any positive $\lambda$, there is an immediate large improvement in adversarial robustness. Next, as $\lambda$ is systematically increased by factors of 10, we do not see a significant change in the accuracy (not under attack). This corroborates Theorem. 4.3, as it demonstrates that regulating knowledge discontinuities (no matter how strongly) is not at odds with minimizing the empirical risk of our model. On the other hand, we also do not see a significant increase in adversarial robustness as $\lambda$ increases. This may imply that we have reached the threshold of adversarial robustness under TextFooler [35]. Specifically, the adversarial attacks generated by TextFooler may not be valid in that they have flipped the ground-truth label. Therefore, we believe that a good $\lambda$ for this particular application should lie somewhere between 0 and $1 \times 10^{-4}$.

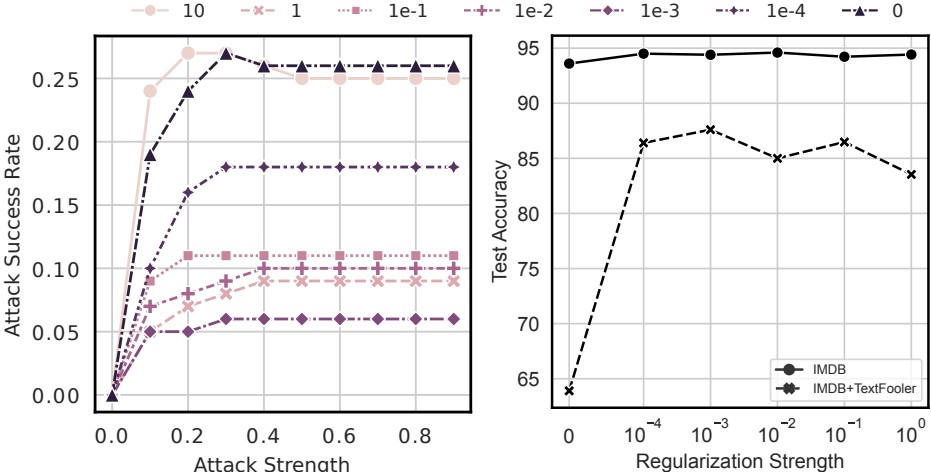

Figure 5: Ablation over the strength of regularization and its effect on the attack strength-attack success rate curves (left). Ablation over the regularization strength (for fixed attack strength $= 0.3$) and its effect on test accuracy (right). We see that moderate regularization significantly improves robustness across all attack strengths. This improvement does not come at the expense of test accuracy. The attack-strength is measured using the minimum angular similarity between the perturbed and original text. Both ablations are done with respect to GPT2 on the IMDB [48] dataset with respect to the TextFooler attack [35].

**Ablation Study of Adversarial Attack Strength (Fig. 5(left)).** For every value of $\lambda$, we also vary the strength of the adversarial attack. The adversarial attack strength is measured through the angular similarity of the embeddings between the original text and the perturbed text. Intuitively, if this constraint is loosened the adversary is allowed to find text that is semantically very different and vice versa. We see that moderate $k$-volatility regulation achieves the best adversarial robustness across all attack strengths.

**Ablation Study of** $(\alpha, \beta)$ In this subsection, we briefly discuss how the $\alpha, \beta$ hyperparameters which determine the shape of the Beta distribution in Alg. 1(KCReg) affect the final performance and robustness of our model on the IMDB dataset. Recall that the shape of the Beta distribution determines the index of the hidden layers we are using the compute the knowledge continuity. Thus, they are crucial in determining the behavior of our regularizer.

We finetune {BERT, T5, GPT2} models on the IMDB dataset with the hyperparameters described in the next subsection. The results are displayed in Table 3. Across all models we observe a decrease in robustness for $\alpha = 1, \beta = 2$. These values correspond to a right-skewed distribution which places high sampling probability on the earlier (closer to the input) hidden layers. Intuitively, perturbations in the early layers should correspond to proportional textual perturbations in the input text. Pure textual perturbations with respect to some metric like the Levenshtein distance should be only loosely if not completely (un)correlated with the actual labels of these inputs. Therefore, enforcing knowledge continuity with respect to this metric should not see increase robustness. Moreover, we also observe a larger decrease in accuracy (not under attack) with the same parameters. This suggests that maintaining this sort of knowledge continuity in the earlier layers is harder to converge on and there may be a "push-and-pull" behavior between optimizing knowledge continuity and accuracy (not under attack). Surprisingly, we observe no significant difference between the other $\alpha, \beta$ values shown in the table.

We did not formally benchmark other configurations of $\alpha, \beta$ such as increasing their magnitude to impose a sharper distribution. ***Anecdotally, during training, we noticed that using these sharper distributions both significantly slowed the model's convergence and decreased the model's accuracy (not under attack). It could be that though knowledge continuity itself is a local property and the enforcement of this local property requires change on a global scale. In other words, one cannot simply reduce the knowledge discontinuities or uniformly converge with respect to one layer without participation from other layers.*** The extent to which other layers are involved in the regularization of a specific one is an interesting question that we leave for future research.

| Model | IMDB | IMDB$_{TF}$ |
|---|---|---|
| BERT$_{BASE}$ | 93.6 | 47.9 |
| BERT$_{BASE}$+Reg$_{(2,1)}$ | **94.8** | **75.1** |
| BERT$_{BASE}$+Reg$_{(2,2)}$ | 89.2 | 74.1 |
| BERT$_{BASE}$+Reg$_{(1,2)}$ | 87.0 | 68.2 |
| GPT2 | 93.6 | 63.9 |
| GPT2+Reg$_{(2,1)}$ | 94.6 | 85.0 |
| GPT2+Reg$_{(2,2)}$ | **94.9** | **87.8** |
| GPT2+Reg$_{(1,2)}$ | 93.1 | 84.9 |
| T5$_{BASE}$ | 93.7 | 53.9 |
| T5$_{BASE}$+Reg$_{(2,1)}$ | **95.0** | 88.9 |
| T5$_{BASE}$+Reg$_{(2,2)}$ | 94.9 | **89.3** |
| T5$_{BASE}$+Reg$_{(1,2)}$ | 94.6 | 88.1 |

Table 3: We train finetune {BERT, T5, GPT2} using knowledge continuity regularization, as described in Alg. 1(KCREG). We varied the $\alpha, \beta$ hyperparameters for the Beta distribution as to determine the effect of these parameters on model performance and robustness. The rows of the table are labeled with the format: Model+Reg$_{(\alpha,\beta)}$. The bolded entries of the table correspond to the best performing metrics out of the knowledge continuity regulated models.

| Hyperparameter | Value |
|---|---|
| Optimizer | Adam |
| Adam $\beta_1$ | 0.9 |
| Adam $\beta_2$ | 0.999 |
| Adam $\epsilon$ | $1 \times 10^{-8}$ |
| Max Gradient Norm | 1.0 |
| Learning Rate Scheduler | Linear |
| Epochs | 20 |
| Batch Size | 32 |
| Learning Rate | $5 \times 10^{-5}$ |
| Weight Decay | $1 \times 10^{-9}$ |

Table 4: Training hyperparameters and optimizer configurations for finetuning models {BERT, GPT2, T5} on IMDB without any form of regularization or adversarial training.

## G.5    Training Details

In this section, we describe in detail the training objectives, procedures, algorithms, and hyperparmeters that we used in the main text and further experiments done in the appendix.

**Brute-Force Adversarial Training.** For all models undergoing adversarial training, we first finetune the model against the training set. Then, attack it using the TextFooler [35] algorithm with examples from the training set. After the attacks are concluded, we then incorporate the text of successful adversarial attacks back into the training set and proceed to finetune again. This procedure iteratively continues. For the sake of computational efficiency, for all models we applied this procedure once. The parameters we are using during the adversarial attack is the same hyperparameters we actually use at test-time. Specifically, we impose a query budget of 300 queries.

**Plain Finetuning on IMDB.** The IMDB dataset consist of 50,000 examples with 25,000 for training and 25,000 for testing. We split the test set 40%-60% to create a validation and test set of 10,000 and 15,000 examples, respectively. Examples were sampled uniformly at random during the splitting process. Since adversarial attacks were costly, we uniformly subsampled 5,000 examples from this 15,000 to benchmark robustness in the experiments related to the regularizer. However, for the experiments estimating the knowledge vulnerability score, we performed adversarial attacks on all 15,000 datapoints in the test set. We found no significant difference between robustness estimation on this 5,000 subsample versus and the entire 15,000 dataset.

We train all models using the hyperparameter and optimizer configurations shown in Table 4.

**Knowledge Discontinuity Regulation on IMDB.** To enforce the knowledge discontinuity on IMDB, we use a constant $\lambda = 1 \times 10^{-2}$ for all models. As shown in Table 3, we varied $\alpha, \beta \in \{1, 2\} \times \{1, 2\}$ and displayed the best models in terms of robustness in Table 1 in the main text. We train all models for 50 epochs. Other than that all the other hyperparameters and optimizer configurations are the same as regular finetuning (see Table 4).

**Knowledge Discontinuity Regulation on ANLI.** Optimizing over the ANLI dataset was significantly harder than on IMDB. As a result, for each model class {BERT, GPT2, T5} we performed a quick hyperparameter search over $\lambda$ ($1 \times 10^{-4}$), the learning rate ($5 \times 10^{-5}$), and weight decay ($1 \times 10^{-9}$) fixing the parameterization of the Beta distribution to be the best values on the IMDB dataset. That is, for T5: $\alpha = 2, \beta = 1$; BERT-Base-Uncased: $\alpha = 2, \beta = 1$; GPT2: $\alpha = 2, \beta = 2$.

**ALUM on IMDB and ANLI.** We train all ALUM models for 50 epochs (the same as knowledge discontinuity regularized models). For hyperpararmeters specific to the ALUM algorithm we choose all of the same ones as its authors, [43], with the exception of $\alpha$ (analogous to the $\lambda$ in our algorithm, essentially the weight put on the virtual adversarial training loss term). The authors of the original paper choose $\alpha = 10$. We, however, found that this applied to finetuning does not converge at all. Thus, with a rough grid search in the parameter space we found $\alpha = 1 \times 10^{-3}$ to be the best with respect to both performance and robustness.

We keep the same hyperparameters on ANLI, however, we impose early stopping during the training process. That is, we choose the best model with respect to its performance on the **dev** set.

## H   Certifying Robustness at Test-Time

Herein, we present a certification algorithm using Thm. 4.1 and our Monte-Carlo estimate of $k$-volatility 1(ESTKVOL). Our algorithm (shown in Alg. 2) is based on the work of [12]. We upperbound the $k$-volatility by bootstrapping a $1 - \alpha$ confidence interval. Then, directly apply Thm. 4.1 using the 0-1 loss function. Thus, Cor. H.1 follows. We emphasize here that this certification algorithm may not be *directly* informative, especially in the discrete/non-metrizable setting, unless we have an inverse map from the representation space back to the input space. This is discussed further in [82]. Nonetheless, it can be used as a method to verify whether or not certain intervention techniques are successful before deploying them in the wild.

**Corollary H.1.** *Let $A = \{(x_i, y_i)_{i=1}^n\}$ and $A' = \{(x', y') \in \mathcal{X} \times \mathcal{Y} : \mathbb{E}_{(x,y)\sim\mathcal{D}_{\mathcal{X},\mathcal{Y}}}\Delta\mathcal{L}_f^{(x,y)}(x', y') > \eta\}$.*

$(x,y)\in A$

*Then, with probability $1 - \alpha$, the output of Alg. 2 bounds $\mathbb{P}[A' | d_j(f^j(x), f^j(A))]$ where $\mathcal{L}$ is the 0-1 loss.*

---

**Algorithm 2** Certifying robustness of a metric decomposable function $f$ with respect to one hidden representation using Alg. 1(ESTKVOL) and Thm. 4.1.

---

**procedure** CERTIFY($f, \{(x_i, y_i)\}_{i=1}^n, k, j, \alpha, \delta, \eta$)
    Let $\mathcal{L}$ be the 0-1 loss function
    $\epsilon_U \leftarrow$ UPPERCONFBOUND($f, \mathcal{L}, \{(x_i, y_i)\}_{i=1}^n, k, j, \alpha$)
    $B \leftarrow \max_{1\le a,b\le n} d_j(f^j(x_a), f^j(x_b))$
    $V \leftarrow \eta\left(1 - \exp\left(-2/B^2 \left(\delta - B\sqrt{\frac{1}{2}\log 2n}\right)^2\right)\right)$
    **return** CLIP($1 - \epsilon_U\delta/V, 0, 1$)
**procedure** UPPERCONFBOUND($f, \mathcal{L}, \{(x_i, y_i)\}_{i=1}^n, k, j, \alpha$)
    $U \leftarrow \mathbf{0}_k$
    **for** $i \leftarrow 1 \dots k$ **do**
        $S \leftarrow$ sample w/ replacement $n$ points from $\{(x_i, y_i)\}_{i=1}^n$
        $U_i \leftarrow$ ESTKVOL($S, \mathcal{L}, f, j$)
    **return** $\frac{1}{k}\sum_{\ell=1}^k U_k + \Phi^{-1}(\alpha)\text{std}(U)/\sqrt{k}$

---

Along these lines, we apply our certification algorithm to our regularized models to verify that the certified robustness has indeed improved. These results are shown in Fig. 6.

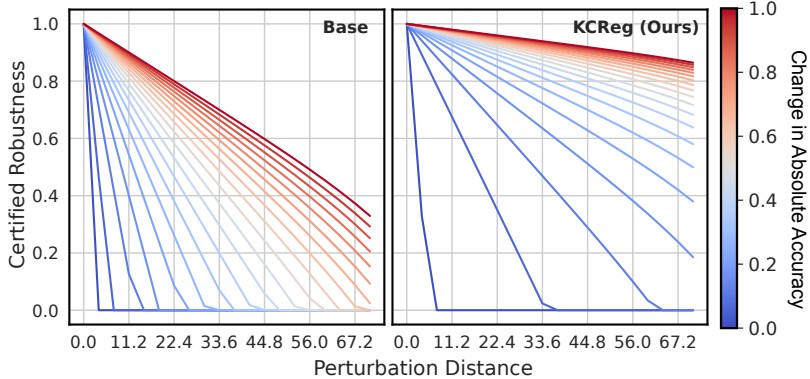

Figure 6: Certification of robustness for GPT2, layer=6. We apply Alg. 2 to certify robustness of the model before and after regularization with Alg. 1(KDREG). Each line corresponds to the change in absolute accuracy for a set of examples to be considered non-robust. The *y*-axis corresponds to the certified probability measure of the set of non-robust examples under this criterion and the *x*-axis corresponds to the maximum perturbation distance in the representation space.

## I  Broader Impacts

This contribution is concerned with robust deep learning models. As deep learning becomes ubiquitous as the primary method for creating artificial intelligence, their applications in increasingly critical areas to the lay and corporations alike demand not only both high inferential accuracy and confidence but also safety and trustworthiness guarantees. Robustness addresses this latter point. More specifically, our contribution unifies separate robustness efforts from continuous and discrete domains.

## J  Reproducibility

All of our experiments were conducted on four NVIDIA RTX A6000 GPUs as well as four NVIDIA Quadro RTX 6000 GPUs. The rest of our codebase including implementations of the algorithms and figures described in the manuscript can be found at https://github.com/alansun17904/kc.

## K  Limitations

The certification guarantees of our definition knowledge continuity is a probabilistic one. Specifically, this randomness is over the data distribution. However, this does not protect against out-of-distribution attacks that plague large language models such as [72, 91]. More work is needed to yield deterministic results that do not become vacuous in discrete settings. As mentioned in Section 4.4, our expressiveness bounds only apply under little restrictions to the metric decompositions of the estimator $f$. Though we see some empirical verification for this in Appendix G, it remains unclear whether or not we can tighten these bounds.

