# OpenReview forum: "Achieving Domain-Independent Certified Robustness via Knowledge Continuity"
_NeurIPS.cc/2024/Conference — NeurIPS 2024 poster_

### Official Review · Reviewer_7Euo · 2024-07-09

**Soundness:** 3
**Presentation:** 3
**Contribution:** 3
**Rating:** 7
**Confidence:** 2

**Summary:**

In this work, the authors propose a novel definition of "continuity": knowledge continuity, which measures the continuity of a model with respect to its hidden representations instead of the inputs. The knowledge continuity can be used to certify adversarial robustness across domain modality. Then, they provide theorems of robustness certification based on the new definition of continuity and demonstrate that in some cases the knowledge continuity implies Lipschitz continuity. Further, they prove that knowledge continuity doesn't affect the universal approximation property of neural networks, which means constraining a network to be knowledge-continuous may not hinder its expressiveness. Lastly, the authors provide some applications of knowledge continuity in improving robustness.

**Strengths:**

* The novel definition of continuity is more general and can be used in discrete domains such as natural language.
* The authors provide solid and interesting theories to support the value of knowledge continuity.
* Some applications are provided to further prove their new definition's effectiveness.

**Weaknesses:**

Some typos:
* In Line 193, delete "with respect to a function $f$"?
* In Line 218, $P_X[A]$ -> $P_{X\times Y}[A]$.

**Questions:**

In this work, the authors decompose the network $f^k$ as $h^k\circ h^{k-1}\circ \dots \circ h^1\circ h^0$. However, some architectures such as ResNet and transformers, have residual structures. So, how can we decompose these networks as above?

**Limitations:**

As the authors said, this work provides some probability results that don't protect against OOD attacks. More deterministic results are needed in future studies.

---

> ### Author Rebuttal · Authors · 2024-08-02
>
> Dear Reviewer `7Euo`,
>
> Thank you for your review. We appreciate your recognition of the theoretical depth of our work, as well as its applications. We would like to address your concerns as follows:
>
> > Some typos
>
> Thank you for pointing these out! They will be fixed in the final manuscript.
>
> > In this work, the authors decompose the network $f^k$ as $h^k \circ h^{k-1} \circ \ldots \circ h^1 \circ h^0$. However, some architectures such as ResNet and transformers have residual structures. So, how can we decompose these networks above?
>
> There are many methods to decompose such structure. Let us consider two fully-connected layers $A, B$ such that $x \to A(x) \to B(A(x)) \to B(A(x)) + x$. Here, the input $x$ forms a residual connection with the output of $B\circ A$. For your reference, here we present two distinct methods.
>
> In the simplest case, we can treat the residual block as a whole as a metric decomposition: $x \to h(x)$ where $h(x) = B(A(x)) + x$. This is the approach we use in practice when dealing with the residual connections. Moreover, this is the standard way to count layers in computer vision and natrual language processing.
>
> On the other hand, if we wish to decompose the residual block itself. Define
> \begin{gather}
>     A' : x\mapsto (A(x), x), \\\\
>     B' : x\mapsto (B(x), x), \\\\
>     x' : (x,y) \mapsto (x+y,y).
> \end{gather}
> It follows that $x' \circ B' \circ A'$ is equal to our residual block. Moreover, the image of each function in this decomposition: $x', B', A'$ forms a metric space. Here, the metric space is with respect to the quotient space where $(a,a') \sim (b,b')$ if and only if $a=b$. In this way, we also recover the same subspace structure as the intermediate operations in the residual block. In other words, we duplicate the input vector, concatenate them together, and only operate on the first vector.
>
> This example and others are included in Appendix A (Lines 570-596). *We want to emphasize that the non-uniqueness of the metric decompositions does not affect the validity of our theoretical results, as we only assume that such a decomposition exists.*
>
> Thanks for your time, and we hope that these clarifications will boost your confidence in our work. We would be happy to discuss any remaining concerns.
>
> Best,
>
> Authors

---

> > ### Comment · Reviewer_7Euo · 2024-08-13
> > **Thanks for your reply**
> >
> > The authors' reply has answered my questions. And I keep the score unchanged.

---

### Official Review · Reviewer_Cz6j · 2024-07-11

**Soundness:** 4
**Presentation:** 4
**Contribution:** 3
**Rating:** 7
**Confidence:** 3

**Summary:**

This paper proposes a generalization of adversarial robustness and certified robustness for continuous, discrete, or non-metrizable input domains. The authors introduce _knowledge continuity_, a novel definition inspired by Lipschitz continuity which aims to certify the robustness of neural networks across input domains (such as continuous and discrete domains in vision and language, respectively). While existing approaches that aim to certify robustness (e.g., Lipschitz continuity) lie in the continuous domain with norm and distribution-dependent guarantees, the new proposed framework proposes certification guarantees that depend only on the loss function and the intermediate learned metric spaces of the neural network.

The current definition of robustness considers the stability of a model's performance with respect to its perceived knowledge of input-output relationships. However, the authors argue that robustness is better achieved by focusing on the variability of a model's loss with respect to its hidden representations, rather than imposing arbitrary metrics on its inputs and outputs.

To summarize the authors' contributions:
- They introduce _knowledge continuity_, a new concept that frames robustness as variability of a model's loss with respect to its hidden representations.
-  They show theoretically that knowledge leads to certified robustness guarantees that generalize across modalities (continuous, discrete, and non-metrizable), and show that this new robustness definition does not come at the expense of inferential performance.
- They present several practical applications of knowledge continuity such as using it  to train more robust models and to identify problematic hidden layers.

**Strengths:**

- The paper is clear and very well written.
- The problem, i.e. defining a robustness framework for discrete or non-metrizable input domains, is very interesting and important as language models become more ubiquitous.
- The new framework seems to generalize previous definitions of robustness, for example, robustness via Lipschitz continuity becomes a special case of knowledge continuity.
- The authors show that their new definition of robustness
- The authors propose experiments to demonstrate their approach on language tasks.

**Weaknesses:**

- While the method seems to be useful for robustness in the context of discrete input domains such as language, the new framework also seems to inherit drawbacks of current certified methods, namely Lipschitz continuity and randomized smoothing:
  - Obtaining robustness from Lipschitz continuity requires having knowledge of the architecture of the network, and constraining the architecture to fit the definition; an example of this is that transformers are not _globally_ Lipschitz continuous, and thus certified robustness for transformer architectures via Lipschitz continuity is difficult. While knowledge continuity could in theoretically be used on any architecture where the hidden layers are metric spaces, I wonder if specific architectures might lead to vacius bounds. Similarly, quantized networks would not fall under the definition of having hidden layers that are metric spaces .
  - As randomized smoothing, the robustness derived from knowledge continuity is not deterministic. This could lead to a high computation of robustness accuracy and certified accuracy.
- The implementation of knowledge continuity is not very clear, I understand that the authors have proposed a regularization scheme via Algorithm 1 for "Estimating knowledge continuity". Algorithm 1 estimates the the expected value that defines knowledge continuity, is this approximation accurate and how does the accuracy of the approximation affect the training speed?
- While the authors have shown that there is theoretically no tradeoff between the robustness of knowledge continuity and natural accuracy, this does not mean that there is no tradeoff in practice. It would be interesting to see the tradeoff between between the strength of the attacks and the strength of the regularization, which seems to be missing from the paper.
- The authors proposed a certified robustness approach, but no certification algorithm was provided. How to compute the certified accuracy with knowledge continuity. Given the non-deterministic approach of their framework, some statistical testing, as in randomized smoothing, may be necessary.

**Questions:**

See Weaknesses.

**Limitations:**

See Weaknesses.

---

> ### Author Rebuttal · Authors · 2024-08-05
>
> Dear Reviewer `Cz6j`,
>
> Thank you for your valuable feedback. We are glad you appreciated the timeliness of our work and its theoretical soundness. We hear your concerns, and hope to clarify the following:
>
> > While knowledge continuity could in theoretically be used on any architecture where the hidden layers are metric spaces, I wonder if specific architectures might lead to vacuous bounds.
>
> We considered this, but determined it did not detract from our work:
>
> Firstly, we disambiguate *architecture* and *metric decomposition*. For a given architecture, there could exist many possible metric decompositions, as a decomposition is dependent not only on the topology induced by the metric but also the metric itself (see Def. 1). For example, consider a fully-connected network whose hidden representations are endowed with either the $\ell_1$ or $\ell_2$ metric; we would consider these to be distinct decompositions even though the architecture is the same.
>
> You are correct in pointing out that there exists decompositions that yield trivial bounds. For example, consider the discrete metric ($d(x,y) = c$ if and only if $x=y$ and $0$ otherwise for some $c \in [0,\infty]$). As $c\to\infty$ knowledge continuity will approach 0 almost everywhere, assuming our loss is finite almost everywhere.
>
> In general, we find these decompositions to be degenerate and unreasonable in practice since they also trivialize robustness. For example, if $c = \infty$, robustness is inherently not well-defined since any perturbation would lead to an infinitely different point. Thus, it would make sense for the bounds to become vacuous anyways.
>
> We conjecture that only metric spaces with infinite Hausdorff dimension behave this way (see Conjecture 4.5, Lines 275-278). Thus, any metric-decomposable neural architecture whose computation is tractable in practice, will lead a "natural metric decomposition" that has non-trivial bounds.
>
> Happy to discuss this more during the discussion phase.
>
> > quantized networks would not fall under the definition of having hidden layers that are metric spaces
>
> One could equip the hidden layers of any quantized network the Hamming distance as its metric or use the $\ell_p$ distances. The choice that makes sense depends on the coarseness of the quantization being performed.
>
> > robustness derived from knowledge continuity is not deterministic. This could lead to a high computation of robustness accuracy and certified accuracy.
>
> Though it would be nice to have deterministic bounds, one of the key insights of our work is that the stochasticity of our framework is necessary for generalizability across different domains.
>
> In continuous domains such as computer vision, the map of the input is dense in any of the metric decompositions. For example, consider a model $f: \mathbb{R}^n \to \mathbb{R}^m$. For any $x, a \in \mathbb{R}^n$ there exists some $x' \in \mathbb{R}^n$ such that $f(x') = f(x) + a$. The same cannot be said for most discrete or non-metrizable applications. Especially in natural language processing, where the correspondence between the learned representation space and the input is provably sparse. This means that any deterministic bound which depends only on the metric in the hidden space cannot capture this nuance.
>
> In addition, we emphasize that for the notion of robustness to generalize across domains, it cannot be task invariant (see Lines 37-40). This necessitates dependence on both the input $x$ and its corresponding label $y$. Thus, we believe that any meaningful domain-independent certification must depend on the data distribution.
>
> We appreciate the feedback and this discussion will be added in the final manuscript.
>
> > Accuracy of Algorithm 1 and effect on training speed?
>
> We use a Monte Carlo approach to implement knowledge continuity: directly calculating the expression inside the expectation. We demonstrate that Alg. 1 is indeed accurate, as it is an unbiased estimator for knowledge continuity (see Proposition G.1). Further, we bound the number of samples needed to compute an arbitrarily accurate approximation of knowledge continuity in Proposition G.2 using a median-of-means approach. In practice, we find that this naive algorithm performs quite well and converges faster than methods of regularization such as ALUM or virtual adversarial training (see Lines 339-340).
>
> > tradeoff between between the strength of the attacks and the strength of the regularization
>
> We run ablations over the strength of the regularization and observe the tradeoffs between test accuracy and accuracy under adversarial attack in Fig. 7 (on Pg. 29). We find that even a small amount of regularization dramatically boosts robustness. And as the magnitude of regularization increases, test accuracy is mostly uncompromised.
>
> We run additional ablations over the adversarial-attack strength and find that our aforementioned results are consistent. For more detail, please see Fig. 3 in the attached pdf in the global response. These results will also be included in the final manuscript.
>
> > no certification algorithm was provided...Given the non-deterministic approach of their framework, some statistical testing, as in randomized smoothing, may be necessary.
>
> Taking inspiration from randomized smoothing as you suggested, we propose an algorithm to compute the certified robustness. This is shown in Alg. 3 in the global response. First, we upper-bound the knowledge continuity by bootstrapping a one-sided confidence interval similar to randomized smoothing, then directly apply Theorem 4.1. The proof of correctness for this algorithm follows as a corollary from Theorem 4.1 and an application of the central-limit theorem. Detailed results are shown in Fig. 2 in the attached global response.
>
> Thanks again for the detailed review. We hope that we addressed all of your concerns, and would be happy to discuss more.
>
> Best,
>
> Authors

---

> > ### Comment · Reviewer_Cz6j · 2024-08-10
> >
> > Thank you for the rebuttal, I think adding the algorithm to compute the certified robustness strengthens the paper. I am in favor of accepting this paper at the conference and have raised my score accordingly.

---

### Official Review · Reviewer_nGPb · 2024-07-17

**Soundness:** 3
**Presentation:** 2
**Contribution:** 2
**Rating:** 5
**Confidence:** 3

**Summary:**

This paper proposes a new concept inspired by Lipschitz continuity, aimed at certifying the robustness of neural networks across different input domains, such as continuous and discrete domains in vision and language. This certification guarantees based on the loss function and intermediate learned metric spaces, independent of domain modality, norms, and distribution. In this paper, the authors define knowledge continuity as the stability of a model's performance to its perceived knowledge of input-output relations. This approach focuses on the variability of a model's loss to its hidden representations rather than imposing arbitrary metrics on inputs and outputs.

**Strengths:**

- The paper generalizes the Lipschitz continuity concept to continuous and discrete domains.
- The paper provides definitions and proofs for knowledge continuity.
- By providing a domain-independent approach to certified robustness.

**Weaknesses:**

- The authors claim that their method generalizes across modalities (continuous, discrete, and non-metrizable). However, they have not conducted experiments in other domains. The experiments in Table 1 are limited to the discrete text domain.
- The paper does not provide specific implementation details. Section 5 directly presents the experimental results in Table 1, but the metrics in Table 1 are not clearly defined.
- From the paper "On the Robustness of ChatGPT: An Adversarial and Out-of-distribution Perspective," it is observed that model robustness improves with increased model capacity. It is necessary to demonstrate that the proposed method works on larger models like LLaMA3-70B/Qwen-1.5-70B to prove that the effectiveness of the method is not merely due to limited model capacity.
- The proposed method does not consistently achieve better performance on ANLI. It remains competitive with previous methods rather than showing clear superiority.
- The datasets used for comparison, IMDB and ANLI, focus more on word-level attacks, which are not particularly challenging for current large language models (LLMs). There is a lack of analysis on LLMs.
- It is unclear how much benefit this method provides to bidirectional attention models like BERT and RoBERTa, which already have some error correction capabilities.
- When dealing with out-of-distribution (OOD) data, the distributions themselves are different. It is uncertain whether this method will still be effective under such conditions.

**Questions:**

- See "Weaknesses"
- The equation in line 252 is not numbered and contains a typo.

---

> ### Author Rebuttal · Authors · 2024-08-02
>
> Dear Reviewer `nGPb`,
>
> Thank you for your time and review. We are glad you appreciated some of the key contributions of our paper, especially that our approach is a rigorous, domain-independent extension of Lipschitz continuity. We hear your concerns, and would like to address them here:
>
> > experiments in other domains
>
> To support our claims (Proposition 4.6-8) and generalize Table 1, we run additional experiments on vision tasks across a set of models whose architectures vary from CNN-based to Transformer-based. (ResNet, MobileNet, ViT). We find that regulating knowledge continuity improves adversarial robustness on almost all of these models across attack methods. Please see our global response and Fig. 1 in the attached PDF for details. These results will also be included in the final manuscript.
>
> > The paper does not provide specific implementation details.
>
> We understand this may have been unclear. So, here are some key areas where implementation details can be found:
> * For empirical model analysis, please see Appendices E-G
> * For estimating knowledge continuity and regularization (Def. 2 and 4, Lines 178-179), please see Alg. 1-2 (Lines 794-795).
> * For ablation studies and training details, please see Appendix G.3, G.4
>
> Moreover, we have now added additional ablations over the attack strength, supplementing Table 1. Please see Fig. 3 in the attached global response PDF. Our regression results and implementation are shown in Appendix E, Fig. 2.
>
> We will use the additional page to expand on our experimental methodology in the main text.
>
> > metrics in Table 1
>
> Metrics in Table 1 are all accuracy since the datasets are class-balanced.
>
> > It is necessary to demonstrate that the proposed method works on larger models like LLaMA3-70B/Qwen-1.5-70B to prove that the effectiveness of the method is not merely due to limited model capacity.
>
> We respectfully disagree that these experiments are necessary to validate the claims in our paper.  The main contribution of our paper is our theoretical framework which does not dependent on the capacity of the model. We show in Theorem 4.1, Proposition 4.3-8 that our robustness guarantees do not become vacuous as capacity goes to infinity. Along these lines, our experiments with BERT, GPT2, and T5 demonstrate that this framework is compatible with fundamental architectures used in language modeling that transcend scale. However, we agree that such experiments are important to the applicability of the framework and would form the basis of interesting future work.
>
> > The proposed method does not consistently achieve better performance on ANLI. It remains competitive with previous methods rather than showing clear superiority.
>
> ANLI is a difficult task, even for LLMs [1,2,3]. Thus, we use this task to demonstrate that the performance of our regularization remains competitive and does not significantly degrade the model's performance as task difficulty scales.
>
> > IMDB and ANLI, focus more on word-level attacks, which are not particularly challenging for current large language models (LLMs). There is a lack of analysis on LLMs.
>
> Based on our extensive literature review, word-level adversarial attacks remain effective even as models scale. Please see results from [2] (last bullet point on Pg. 3), [3] (see Fig. 3.9 and Table 3.2), and [4] (see Table 4 and section 4.3). However, we agree that an important future goal is to explain and regulate complex phenomena such as jailbreaking, hallucination, bias, etc.
>
> To this end, a key contribution of our method is that our bounds do not depend on the type of perturbation on the input, only its *task-dependent* semantic distance. Thus, we expect our framework to generalize to these more complex tasks and adversaries. We leave these topics for future work.
>
> > It is unclear how much benefit this method provides to bidirectional attention models
>
> We have included results for BERT in Table 1. Moreover, T5 also includes bidirectional attention modules for which our regularization method performs well.
>
> > OOD data
>
> We acknowledge in our limitations section that our method does not protect against OOD attacks (see Lines 345-346). As this is not the focus of the paper, we leave the extension of our framework for future work. As in-distribution adversarial attacks are pervasive [5,6,7] in a host of domains, we believe that this limitation does not detract from the timeliness nor the contribution of this work. We would love to hear any of your suggestions on how to extend this work.
>
> > line 252 typo
>
> Thanks for catching this error. This will be fixed in the final manuscript.
>
> Thanks again for your review, and we hope that these clarifications addressed your concerns. We would be happy to discuss further.
>
> Best,
>
> Authors
>
> [1] Chowdhery, Aakanksha, et al. "Palm: Scaling language modeling with pathways." Journal of Machine Learning Research 24.240 (2023): 1-113.
>
> [2] Ye, Junjie, et al. "A comprehensive capability analysis of gpt-3 and gpt-3.5 series models." arXiv preprint arXiv:2303.10420 (2023).
>
> [3] Brown, Tom, et al. "Language models are few-shot learners." Advances in neural information processing systems 33 (2020): 1877-1901.
>
> [4] Wang, Jindong, et al. "On the robustness of chatgpt: An adversarial and out-of-distribution perspective." arXiv preprint arXiv:2302.12095 (2023).
>
> [5] Chao, Patrick, et al. ‘Jailbreaking Black Box Large Language Models in Twenty Queries’. R0-FoMo:Robustness of Few-Shot and Zero-Shot Learning in Large Foundation Models, 2023.
>
> [6] Qi, Xiangyu, et al. ‘Fine-Tuning Aligned Language Models Compromises Safety, Even When Users Do Not Intend To!’ The Twelfth International Conference on Learning Representations, 2024.
>
> [7] Zou, Andy, et al. "Universal and transferable adversarial attacks on aligned language models." arXiv preprint arXiv:2307.15043 (2023).

---

> > ### Comment · Reviewer_nGPb · 2024-08-11
> >
> > Thank you for your detailed responses. I appreciate your clarifications, and most of my concerns have been addressed. I've raised the rating score accordingly.

---

### Official Review · Reviewer_88sL · 2024-07-25

**Soundness:** 3
**Presentation:** 2
**Contribution:** 3
**Rating:** 7
**Confidence:** 2

**Summary:**

This paper proposes "knowledge continuity", a metric inspired by the Lipschitz continuity that aims to certify the robustness of neural networks. Compared to existing methods, the proposed metric yields certification guarantees that only depend on loss functions and the intermediate learned metric spaces of neural networks, which are independent of domain modality, norms, and data distribution. The authors further show that the knowledge continuity is not conflicted with the expressiveness of a model class. Several practical applications are provided.

**Strengths:**

- The metric seems to be novel, has an elegant form and seems to be able to provide good explanation of robustness.
- The proposed metric is supported by a good amount of theoretical proofs, stating that improving the knowledge continuity does not be conflicted with the representation power of the neural networks.
- practical results indicate that regularizing the metric can help the robustness.

**Weaknesses:**

- A lot of details of the empirical results are deferred to the appendix which makes me hard to understand what are the experimental settings. The authors should consider reorganize the paper - right now it seems that the paper is actually compressed from a longer version (say journal) but not dedicated to the conference. For example, the regression test seems to be very interesting but there is no any actual result shown in the main text.
- Some numbers are wrongly bolded in the table (or the meaning of bolded results is not clearly conveyed) since they are not the highest number.

**Questions:**

I don't have questions.

**Limitations:**

Yes, the authors have discussed the limitation in the main text.

---

> ### Author Rebuttal · Authors · 2024-08-02
>
> Dear Reviewer `88sL`,
>
> We appreciate your thoughtful feedback. We are glad you noticed the elegance and novelty of our metric, and appreciated the way our theoretical results tie nicely into practical applications. We address your concerns as follows:
>
> > A lot of details of the empirical results are deferred to the appendix which makes me hard to understand what are the experimental settings.
>
> We want to emphasize that the focus of our paper is to introduce the theoretical framework of knowledge continuity and its implications on certified robustness. As a result, the practical experiments are more so a proof-of-concept.
>
> In the final manuscript, we will use the additional page to expand on our experimental methodology: including Alg. 1, Fig. 4, and the detailed regression results (see Fig. 2).
>
> To further clarify the practical applications section, we first devise a simple Monte-Carlo method for estimating the knowledge continuity (Def. 2 and 4, Lines 178-179, Lines 197-198) of a network (see Alg. 1, Lines 794-795). Then, this quantity (for brevity, let's call this KVS) is used in three distinct experiments:
>
> 1. We run a **linear regression using KVS to predict accuracy under adversarial attack.** We find that KVS does in fact predict accuracy under adversarial attack, explaining 35\% of the variance which is statistically significant.
> 2. We finetune a set of models (see Table 1) on the IMDB sentiment classification task and **append KVS directly to the cross-entropy loss function.** We find that **adding this term increases accuracy under adversarial attack without sacrificing inferential performance.** This corroborates our theoretical results in Theorem 4.1 (Lines 218-22) and Proposition 4.3-4 (Lines 247-263).
> 3. Lastly, since KVS is a layer-specific metric, **we measure how KVS changes with network depth and perform a qualitative analysis.** In a nutshell, we find that **models which belong to different model families** (encoder, decoder, encoder-decoder) **have drastically different behavior** with respect to KVS. These plots are shown in Fig. 4 (Pg. 24).
>
> > Some numbers are wrongly bolded in the table (or the meaning of bolded results is not clearly conveyed) since they are not the highest number.
>
> The bold numbers simply correspond to our proposed method within each model family. We understand that our bolding scheme was confusing, and this will be corrected in the final manuscript to only bold the highest numbers in each column.
>
> We hope that these clarifications will boost your confidence in our paper. We would be happy to discuss any remaining concerns.
>
> Thank you for your time.
>
> Best,
>
> Authors

---

### Author Rebuttal · Authors · 2024-08-05

Dear All Reviewers,

Thank you for your time and valuable insights into our work. In particular, we are glad that multiple reviewers appreciated the rigor of our theoretical work, the way our definitions generalize various robustness metrics across domains, and the importance of our practical experiments. We believe our work is especially pertinent as language models become more ubiquitous, as the reviewers pointed out.

## Regularization of CV Models

Following the suggestion of Reviewer `nGPb` and Reviewer `Cz6j`, we have conducted additional empirical studies to strengthen our theoretical results. In the paper, we prove (Lines 285-313) that in continuous domains knowledge continuity is equivalent to Lipschitz continuity and recovers its tight robustness guarantees. To empirically test this, we regularize the knowledge continuity of several vision models and apply two adversarial attacks from [1] and [2]. We see that in most cases regularizing knowledge continuity improves adversarial robustness (see Fig. 1 in the attached pdf).

## A New Empirical Certification Algorithm

Next, we build off of our Monte-Carlo algorithm for estimating knowledge continuity (see Alg. 1, Lines 794-795), certification bound in Theorem 4.1 (Lines 218-221) and present an algorithm to empirically certify robustness. We apply this algorithm on GPT2 before and after regularization and demonstrate that our proposed method of regularization does improve knowledge continuity and results in a tighter certification bound. This algorithm is inspired by randomized smoothing and its proof of correctness follows directly from Theorem 4.1 as well as the central-limit theorem. The algorithm and its results can be found in Alg. 3 and Fig. 2 in the attached pdf, respectively.

```
Algorithm 3: Certifying the robustness of a neural network

Function CERTIFY(f, dataset, k, j, alpha, delta, eta):
1. Define L as the 0-1 loss function
2. Calculate epsilon_U using UPPERCONFBOUND function
3. Find the maximum difference B between f^j(x_a) and f^j(x_b) for all 1<=a, b <= n
4. Calculate V using formula given in Theorem 4.1 (Lines 218-222)
5. Return the clipped value of (1 - epsilon_U * delta / V) between 0 and 1

Function UPPERCONFBOUND(f, L, dataset, k, j, alpha):
1. Initialize U as a zero vector of dimension k
2. For i from 1 to k:
   a. Create a sample S by randomly selecting n points from the dataset with replacement
   b. Calculate U_i using Algorithm 1 (Lines 794-794) with S, L, f, and j
3. Return the mean of U + the std of U multiplied by the inverse normal cdf applied to alpha
```

## Additional Ablation Studies Over Regularization and Attack Strength

Lastly, we perform ablations over the regularization strength and examine its effect on the attack strength-attack success rate curves and test accuracy. We find that moderate regularization significantly improves robustness across all attack strengths and this improvement does not come at the expense of test accuracy (see Fig. 3 in the attached PDF).

All of these results, algorithms, and corresponding proofs will be included in the final manuscript.

We hope that these additional experimental results will improve your confidence in our work. We would be happy to discuss any remaining concerns.

Best,

Authors


[1] Goodfellow, Ian J., et al. ‘Explaining and Harnessing Adversarial Examples’. arXiv [Stat.ML], 2015, http://arxiv.org/abs/1412.6572. arXiv.

[2] Lin, Jiadong, et al. ‘Nesterov Accelerated Gradient and Scale Invariance for Adversarial Attacks’. International Conference on Learning Representations, 2020, https://openreview.net/forum?id=SJlHwkBYDH.

---

### Decision · Program_Chairs · 2024-09-25

**Decision:**

Accept (poster)

**Comment:**

The paper received all positive ratings before the rebuttal. The reviewers think the idea is novel and the empirical results are promising. After the rebuttal, most of the minor concerns have been addressed. The AC made the acceptance recommendation.